# Ribosome demand links transcriptional bursts to protein expression noise

**Sampriti Pal, Upasana Ray, Riddhiman Dhar***

Department of Bioscience and Biotechnology, IIT Kharagpur, Kharagpur, India

## eLife Assessment

This study focuses on a previously reported positive correlation between translational efficiency and protein noise. Using mathematical modeling and analysis of experimental data the authors reach the **valuable** conclusion that this phenomenon arises due to ribosomal demand. While some aspects of the work appear to be **incomplete**, the results have the potential to be of value and interest to the field of gene expression.

**\*For correspondence:**
riddhiman.dhar@iitkgp.ac.in

**Competing interest:** The authors declare that no competing interests exist.

**Abstract** Stochastic variation in protein expression generates phenotypic heterogeneity in a cell population and has an important role in antibiotic persistence, mutation penetrance, tumor growth, and therapy resistance. Studies investigating molecular origins of noise have predominantly focused on the transcription process. However, the noise generated in the transcription process is further modulated by translation. This influences the expression noise at the protein level which eventually determines the extent of phenotypic heterogeneity in a cell population. Studies across different organisms have revealed a positive association between translational efficiency and protein noise. However, the molecular basis of this association has remained unknown. In this work, through stochastic modeling of translation in single mRNA molecules and empirical measurements of protein noise, we show that ribosome demand associated with high translational efficiency in a gene drives the correlation between translational efficiency and protein noise. We also show that this correlation is present only in genes with bursty transcription. Thus, our work reveals the molecular basis of how coding sequence of genes, along with their promoters, can regulate noise. These findings have important implications for investigating protein noise and phenotypic heterogeneity across biological systems.

## Introduction

Genetically identical cells often show variation in gene expression under identical environmental conditions. Stochastic variation in gene expression, termed expression noise, generates phenotypic heterogeneity which has important implications for many biological processes. Gene expression noise has been associated with persistence of microbial cells in antibiotics (*Balaban et al., 2004*; *Wakamoto et al., 2013*), incomplete mutation penetrance (*Raj et al., 2010*; *Burga et al., 2011*; *Eldar et al., 2009*), cellular decision-making (*Lu et al., 2021*; *Coomer et al., 2022*), cancer progression, and anti-cancer therapy resistance (*Nguyen et al., 2016*; *Sharma et al., 2019*; *Emert et al., 2021*). Transcription occurring in bursts (*Golding et al., 2005*; *Cai et al., 2006*; *Raj et al., 2006*), with a promoter transitioning between on- and off-states, is a major source of expression noise, and each promoter has its characteristic burst frequency and burst size (*Silander et al., 2012*; *Newman et al., 2006*; *Bar-Even et al., 2006*; *Suter et al., 2011*; *Dar et al., 2012*). Earlier studies have associated the presence of TATA box motif (*Tirosh et al., 2006*; *Raser and O'Shea, 2004*), occurrence of TFIID and SAGA co-activator complexes with TATA-binding protein (*Ravarani et al., 2016*), nucleosome

occupancy, and histone modifications with expression noise (*Tirosh and Barkai, 2008*; *Choi and Kim, 2009*; *Chen and Zhang, 2016*; *Faure et al., 2017*). In addition, transcription factors and the gene regulatory network play important roles in determining noise (*Donovan et al., 2019*; *Loell et al., 2022*; *Parab et al., 2022*).

The expression noise generated by transcription is modulated by the translation process (*Raj et al., 2006*; *Taniguchi et al., 2010*). Thus, a gene that exhibits high expression noise at the mRNA level may not show high noise at the protein level. Two early studies observed a positive association between translational efficiency and expression noise in bacteria and yeast (*Ozbudak et al., 2002*; *Blake et al., 2003*), where an increase in mean protein expression, engineered by changes in synonymous codons, led to an increase in expression noise. This contrasted with the traditional inverse relationship between mean protein expression and noise observed in empirical measurements of noise in bacteria and yeast (*Silander et al., 2012*; *Newman et al., 2006*; *Bar-Even et al., 2006*). (*Salari et al., 2012*), through a computational analysis of expression noise in yeast, observed an increase in positive correlation between tRNA adaptation index (tAI), a proxy for translational efficiency (*dos Reis et al., 2004*; *Tuller et al., 2010*), and expression noise for genes with tAI values up to 0.55. The correlation, however, decreased thereafter (*Salari et al., 2012*). In addition, a recent study in *Arabidopsis* reported a positive correlation between translational efficiency and expression noise (*Wu et al., 2022*), similar to what had been observed in microbes. Taken together, these studies showed that the translation process impacted protein expression noise in a similar manner across biological systems.

What is the molecular basis of the positive correlation between translational efficiency and protein expression noise? The answer to this question remains unclear, although some mechanisms have been proposed. To achieve a specific protein expression level, a cell can produce many mRNA molecules and translate them at a low rate or can start with a small number of mRNA molecules and translate them at a high rate (*Fraser et al., 2004*). It has been argued that the latter scenario requiring high translation rate may lead to more noise, due to constraints on availability of mRNA, or due to fluctuations in small mRNA numbers (*Fraser et al., 2004*; *Pilpel, 2011*). However, there has only been indirect evidence in support of this hypothesis (*Fraser et al., 2004*). Further, recent studies have shown that the translation process is bursty in nature and can switch between on- and off-states (*Wu et al., 2016*; *Livingston et al., 2023*), just like the transcription process. Whether translational bursting along with transcriptional bursting can explain the correlation between translational efficiency and noise remains unknown.

In this work, we show that fluctuations in mRNA levels, originating from bursty transcription, combined with high translational efficiency can lead to high protein noise. Through stochastic modeling of translation in single mRNA molecules, we uncovered that the extent of demand for the ribosomal machinery needed for translation at the level of individual genes determined how transcriptional noise was translated into protein noise. In addition, we found that the positive association between translational efficiency and protein noise was visible only for genes that showed bursty transcription. Consequently, the ribosome demand model predicted that a reduction in translational efficiency or a departure from bursty transcription to a more uniform rate of transcription would abolish the association between translational efficiency and protein noise. We validated both these predictions through an experimental noise measurement system in yeast, thereby establishing ribosome demand as the molecular link between transcriptional bursts and protein noise. Taken together, our work reveals the molecular basis of how coding sequence of genes can regulate protein noise via modulating the translation rate. These findings have implications for investigating protein noise and phenotypic heterogeneity across biological systems.

## Results

### High protein noise in genes with low transcription and high translation stems from high mRNA noise

We first tested the existing hypothesis which posits that the genes with low transcription and high translation would exhibit more protein expression noise than the genes with high transcription and low translation (*Figure 1A*). Although it has been reported that essential genes in the yeast *Saccharomyces cerevisiae* employ high transcription and low-translation strategy for expression, presumably because of detrimental effects of high noise in these genes (*Fraser et al., 2004*), there is no direct

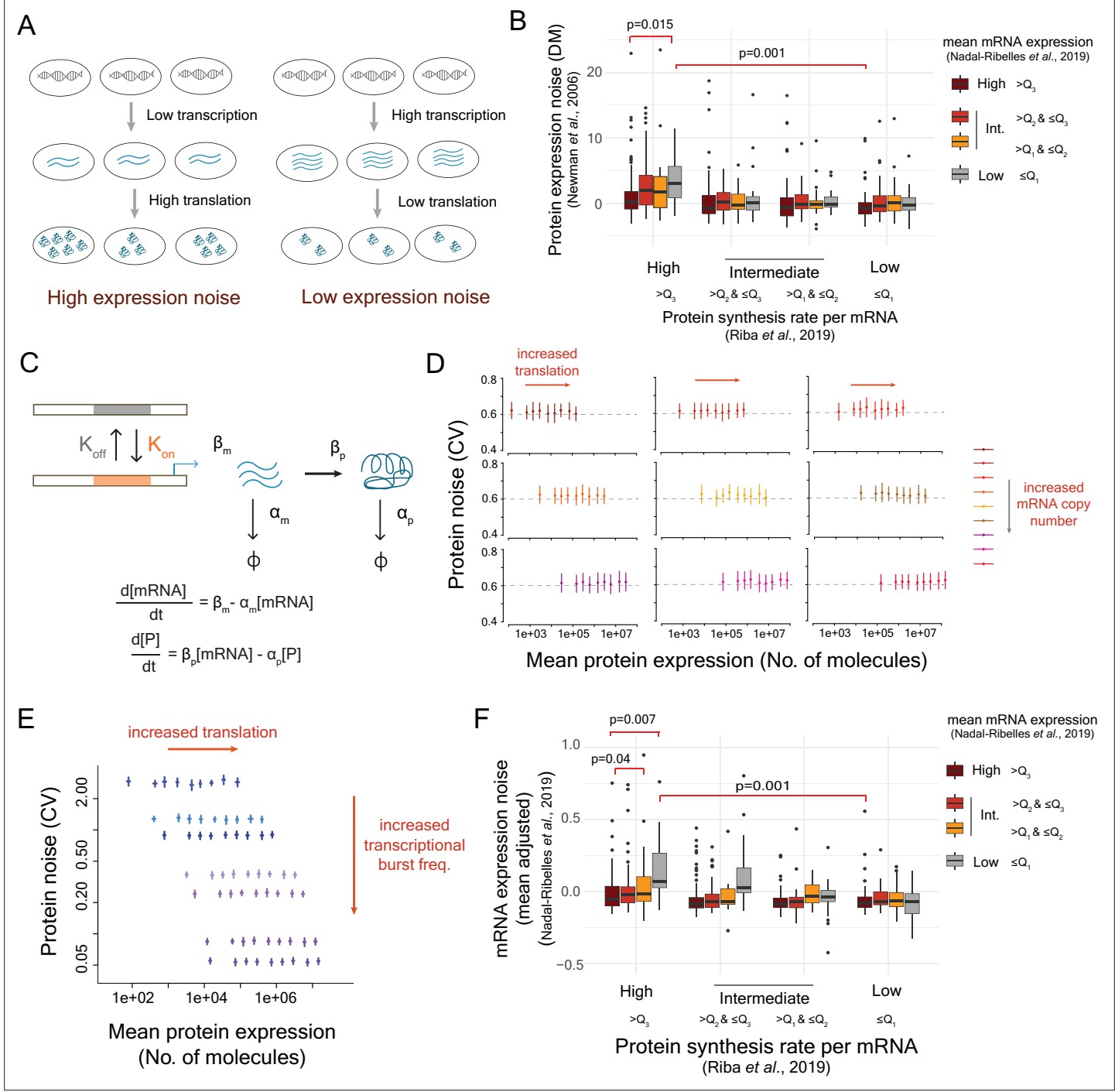

**Figure 1.** High mRNA noise combined with high translational efficiency leads to high protein noise. (**A**) Existing model describing the impact of translational efficiency on protein expression noise. (**B**) Genes were classified into 16 classes according to the quartiles of mean mRNA expression, calculated from *Nadal-Ribelles et al., 2019*, and then by the quartiles of protein synthesis rates per mRNA from *Riba et al., 2019*. The protein noise values for genes in each of the classes were obtained from *Newman et al., 2006*, and the measure distance-to-median (DM) value, as derived in their work, was considered as the measure of noise. (**C**) Two-state model of gene expression with the transition rates $K_{on}$ and $K_{off}$ between transcriptional ON and OFF states was used for stochastic simulations. (**D, E**) Relationship between mean protein expression and protein noise (coefficient of variation, CV) obtained from stochastic modeling using the two-state model. The panel (**D**) describes the results of stochastic simulations obtained at different starting mRNA numbers of a gene to test whether the mRNA expression level of a gene can explain the positive relationship between mean protein expression and protein noise. The panel (**E**) shows the results obtained from stochastic simulations at different transcriptional burst frequencies, but keeping the starting mRNA number of the gene constant. (**F**) Mean-adjusted mRNA expression noise calculated from the single-cell RNA-seq data *Nadal-Ribelles*

*Figure 1 continued on next page*

*Figure 1 continued*

***et al., 2019*** in 16 classes of genes classified according to the quartiles of mean mRNA expression and the quartiles of translational efficiency based on the data on protein synthesis rate per mRNA from ***Riba et al., 2019***. $Q_1$, $Q_2$, and $Q_3$ represent first, second, and third quartiles.

The online version of this article includes the following figure supplement(s) for figure 1:

**Figure supplement 1.** Density distributions of experimental parameters utilized in the present study.

**Figure supplement 2.** Correlation of protein expression noise, distance-to-median (DM) from ***Newman et al., 2006***, (**A**) with mean mRNA expression, calculated from ***Nadal-Ribelles et al., 2019***, and (**B**) with protein synthesis rate per mRNA from ***Riba et al., 2019***.

**Figure supplement 3.** Calculation of mRNA expression noise in yeast and its correlation with mean mRNA expression and protein expression noise.

supportive evidence. To test this hypothesis, we compared protein expression noise of genes showing different levels of transcription and translation in the yeast *S. cerevisiae*. We obtained the mRNA expression data from the study of ***Nadal-Ribelles et al., 2019***, who quantified mRNA expression at the level of single cells in yeast (***Figure 1—figure supplement 1A***). We used the data on protein synthesis rate per mRNA from ***Riba et al., 2019*** as a measure of translational efficiency (***Figure 1— figure supplement 1B***). We obtained the protein expression noise data from ***Newman et al., 2006***, and used their measure of distance-to-median (DM), that is corrected for the dependence of expression noise (coefficient of variation [CV] = standard deviation of protein expression level/mean protein expression) on mean protein expression, as the measure of protein noise. We obtained protein noise values of 2763 genes from this dataset (***Figure 1—figure supplement 1C***).

Mean mRNA expression and translational efficiency individually showed poor correlation with protein noise (***Figure 1—figure supplement 2***). We therefore classified yeast genes based on the quartiles of mean mRNA expression values, and then by the quartiles of translational efficiency, resulting in a total of 16 classes (***Figure 1B***). The group of genes with low mean mRNA expression and high translational efficiency showed significantly higher protein expression noise (DM, median noise = 3.02) compared to the genes with high mRNA expression and low translational efficiency (median protein noise = –0.724; Mann–Whitney *U*-test, p = 0.001; ***Figure 1B***), suggesting that low-mRNA expression and high translational efficiency were indeed associated with high protein noise.

To better understand how low-mRNA expression and high translational efficiency could lead to high protein noise, we employed a two-state model of gene expression (***Peccoud and Ycart, 1995***). Briefly, a gene can switch between on- and off-states with rates $K_{on}$ and $K_{off}$, respectively (***Figure 1C***). In the on-state, the gene is transcribed at a rate $\beta_m$, and the mRNA molecules produced are translated at a rate $\beta_p$. Using stochastic simulations following Gillespie's algorithm (***Gillespie, 1977***), we modeled the changes in concentrations of mRNA and protein molecules over time individually in 1000 cells and further calculated mean protein expression and noise values (CV) (see Methods). In all simulations, we quantified protein expression noise by CV as it was not possible to calculate mean-adjusted noise or DM for a single gene. We modeled the impact of mRNA expression and translational efficiency on expression noise by varying the transcription ($\beta_m$) and the translation ($\beta_p$) rates over a wide range of values (***Figure 1D***). However, we did not see any change in protein noise, either with changes in the translation rate for a specific mean mRNA expression or with changes in the mean mRNA expression (CV = ~0.62, ***Figure 1D***). Further, simulations over a wide range of burst frequencies ($K_{on}$) did not reveal any correlation between translational efficiency and protein noise (***Figure 1E***). These results suggested that the differences in the mRNA levels and the translational efficiency could not explain the differences in noise values between low-mRNA high-translation class compared to the high-mRNA low-translation class.

Next, we tested whether higher heterogeneity in mRNA numbers associated with low mean mRNA expression could explain the higher protein noise in the genes of the low-mRNA high-translation class (***Pilpel, 2011***) from the experimental data. To do so, we derived a measure of mean-adjusted mRNA expression noise (***Parab et al., 2022***) that accounts for dependence of mRNA expression noise on mean mRNA expression, for 5475 genes from yeast single-cell RNA-seq data (***Nadal-Ribelles et al., 2019***; ***Figure 1—figure supplement 3A, B***). Overall, some of the genes with low mean mRNA expression showed high mean-adjusted mRNA expression noise (***Figure 1—figure supplement 3C***). Next, we compared mRNA expression noise of the 16 classes of genes as obtained based on the levels of mean mRNA expression and translational efficiency (***Figure 1B***). The class of genes with low mean mRNA expression and high translational efficiency showed the highest mRNA noise among all classes

(*Figure 1F*). Specifically, this class of genes had significantly higher mRNA noise (median mRNA noise = 0.068) compared to the class with high mRNA level and low translational efficiency (median mRNA noise = –0.082; Mann–Whitney *U*-test, p = 0.001, *Figure 1F*). In general, mean-adjusted protein expression noise showed a moderate correlation with mean-adjusted mRNA noise (Pearson's correlation = 0.44 and Spearman's correlation = 0.29, *Figure 1—figure supplement 3D*). These results suggested that high mRNA noise associated with low transcription, rather than the low mean mRNA level itself, combined with translational efficiency was a good determinant of protein noise.

## Bursty transcription combined with high translational efficiency leads to high protein noise

Based on the above observations, we put forward a new working model that highlighted the influence of mRNA noise in determining protein noise. Noise in mRNA expression is usually associated with bursty transcription, where low burst frequency leads to high mRNA expression noise and an increase in burst frequency gradually lowers mRNA noise as the transcription rate becomes more uniform. Therefore, we hypothesized that among genes that exhibited bursty transcription, the ones with high translational efficiency would show higher protein noise compared to the ones with low translational efficiency (*Figure 2A*).

To test this new model, we estimated the parameters associated with transcriptional bursts such as on- and off-transition rates ($K_{on}$ and $K_{off}$) and transcription rate ($\beta_m$) for >6000 genes in yeast from single-cell RNA-seq data (*Nadal-Ribelles et al., 2019*; *Figure 2—figure supplement 1*), using a maximum-likelihood approach based on a Poisson-Beta model (*Kim and Marioni, 2013*; *Figure 2B*). Genes with low transcriptional burst frequencies showed high mRNA noise (*Figure 2—figure supplement 2*). However, the genes that exhibited high protein noise were associated with high translational efficiency (*Figure 2C*). In addition, we classified genes into 16 classes according to the quartiles of burst frequency and the quartiles of translational efficiency (*Nadal-Ribelles et al., 2019*; *Figure 2D*). Genes with low burst frequency and high translational efficiency showed the highest protein noise among all 16 classes (*Figure 2D*). This class of genes showed a mean protein noise (DM) value of 5.65, whereas the class of genes with high burst frequency and high translational efficiency showed a mean protein noise (DM) value of –0.038 (Mann–Whitney *U*-test, p = 1.22 × 10⁻⁷; *Figure 2D*). These observations confirmed that high translational efficiency, combined with bursty transcription, could lead to high protein noise (*Figure 2D*). Analysis with tAI as the measure of translational efficiency showed a similar trend (*Figure 2—figure supplements 3 and 4*), suggesting that these results were robust to the use of different measures of translational efficiency.

## Bursty translation does not explain the correlation between translational efficiency and protein noise

Even though results from the analysis of earlier experimental data agreed with the new working model, they did not explain how high translational efficiency combined with low burst frequency could lead to high protein noise, and how reducing translational efficiency could lower protein noise. Recent studies have observed that translation, like transcription, also occurs in bursts (*Wu et al., 2016*; *Livingston et al., 2023*). Specifically, (*Livingston et al., 2023*), through single-cell mRNA tracking, measured the parameters of translational bursting and identified the roles of 5′UTR and 5′mRNA cap in modulation of translational burst frequency and burst amplitude.

Therefore, we asked whether incorporating translational bursting into our model could reveal the positive correlation between translational efficiency and protein noise. To do so, we built a TASEP-based model (*Zia et al., 2011*; *Andreev et al., 2018*) of translation, with appropriate modifications, at the level of single mRNA molecules (*Figure 3A*, see Methods). We assumed that an mRNA molecule could transition between on- and off-states (*Livingston et al., 2023*) with the rates $TL_{on}$ and $TL_{off}$, respectively, and in the on-state, translation initiation occurred at the rate of $TL_{init}$ (*Figure 3A*). As multiple ribosomes could translate a single mRNA molecule at the same time, a second translation initiation happened only when the preceding ribosome traversed at least 10 codons, to account for steric interaction between ribosomes (*Steitz, 1969*; *Ingolia et al., 2009*; *Figure 3B*). Several earlier studies have shown the presence of a gradually increasing profile of translational efficiency or ramp, although of different degree, in the 5′ end of coding regions of genes (*Tuller et al., 2010*; *Weinberg et al., 2016*). This meant that the translation speed was lower near the 5′ end of a gene

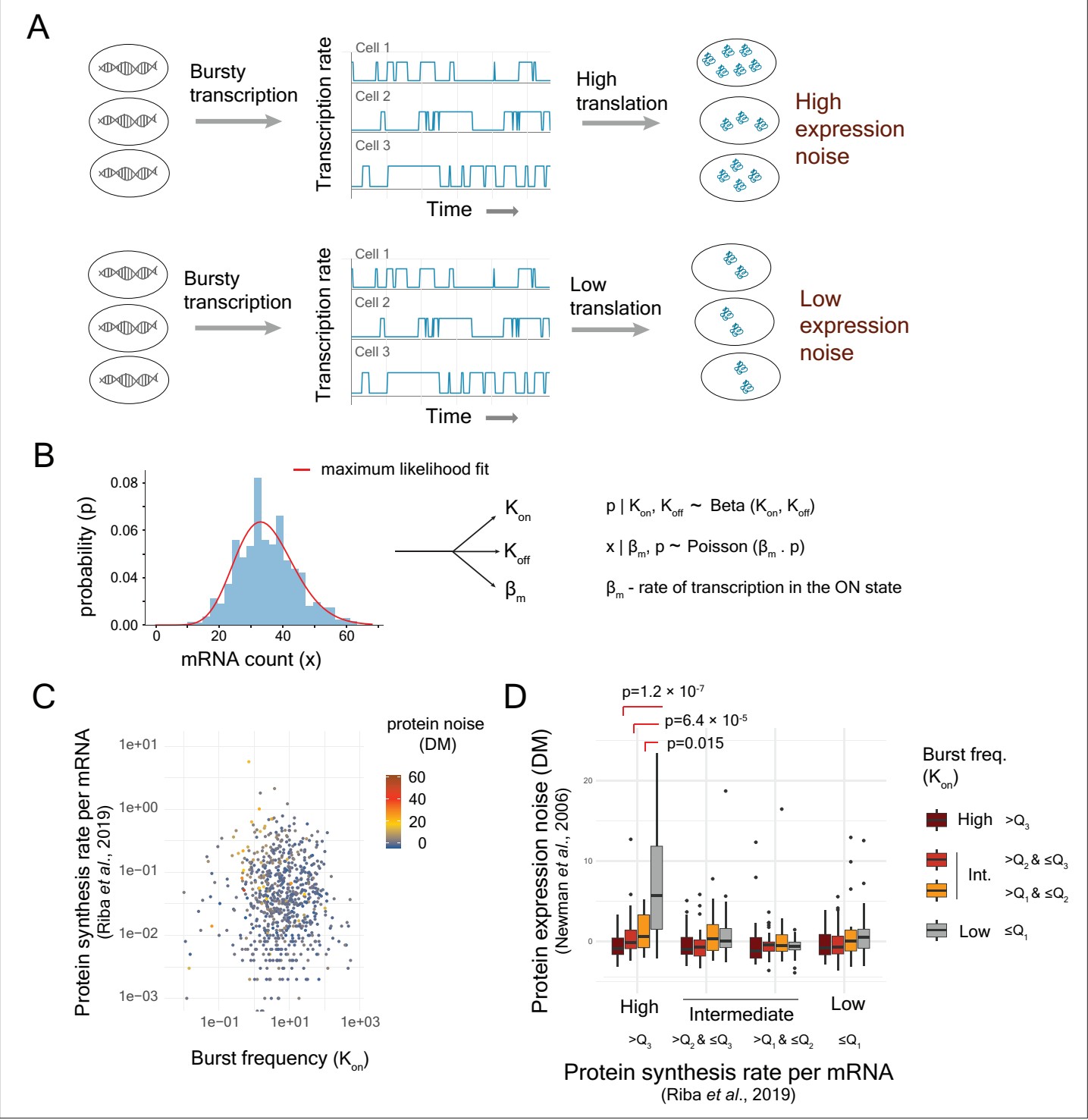

**Figure 2.** Stochastic fluctuation in mRNA expression, originating from transcriptional bursts, combined with high translational efficiency generates high protein noise. (**A**) The new working model postulated that the genes with bursty transcription (low transcriptional burst frequency) and high translational efficiency were likely to exhibit higher protein expression noise compared to the genes with bursty transcription but low translational efficiency. (**B**) Estimation of parameters of two-state model of gene expression from single-cell RNA-seq data as described by *Kim and Marioni, 2013*. (**C**) Protein noise of genes with different levels of transcriptional burst frequencies and translational efficiency, estimated by protein synthesis rates per mRNA (*Nadal-Ribelles et al., 2019*). (**D**) Protein expression noise (distance-to-median [DM] values from *Newman et al., 2006*) of genes classified into 16 classes based on burst frequency and translational efficiency (protein synthesis rate per mRNA; *Riba et al., 2019*). $Q_1$, $Q_2$, and $Q_3$ represent first, second, and third quartiles, respectively.

*Figure 2 continued on next page*

*Figure 2 continued*

The online version of this article includes the following figure supplement(s) for figure 2:

**Figure supplement 1.** Distributions of burst parameters for yeast genes, estimated using the two-state model of transcription, and correlations between them.

**Figure supplement 2.** Relationship between transcriptional burst frequency ($K_{on}$), protein synthesis rate per mRNA (*Riba et al., 2019*), and mean-adjusted mRNA expression noise.

**Figure supplement 3.** Distribution of tRNA adaptation index (tAI) across yeast genes and its correlation with experimentally measured protein synthesis rate per mRNA molecule.

**Figure supplement 4.** Correlation of tAI with burst parameters and its association with protein expression noise.

and the speed gradually increased as a ribosome traversed from 5′ to 3′ end of an mRNA molecule. Taking this into account, we modeled the traversal speed of the ribosomes using a first-order Hill function where the traversal speed ($V$) was dependent on the position of the ribosome in the coding sequence (*Equation 10*; *Figure 3B*). An increase in the value of the parameter $K_{Hill}$ led to an increased traversal time (*Figure 3C*). The speed of traversal has an impact on translation initiation rate, as observed by *Barrington et al., 2023*. This meant that $K_{Hill}$ was also related to the translation initiation rate, $TL_{init}$. Higher translation initiation rate necessitated lower $K_{Hill}$ to ensure faster traversal of the preceding ribosome through the first 10 codons to avoid collision between two successive ribosomes (*Figure 3D*). Similarly, lower $K_{Hill}$ value led to faster traversal of ribosomes through an mRNA molecule, and thereby, permitted a greater number of translation initiation events per unit time (*Figure 3D*). Thus, the translation initiation rate and the ribosome traversal time through an mRNA molecule were interconnected (*Figure 3D*).

We integrated the model of translation with the two-state model of transcription and performed stochastic simulations individually in 1000 cells to estimate mean protein expression and protein noise (CV). To test whether incorporation of the translational bursting could capture the positive correlation between mean protein expression and protein noise, we changed the translational efficiency by varying the translation initiation rate ($TL_{init}$) in our simulations, thereby also affecting the ribosome traversal time, while keeping all other parameters constant. For all transcriptional burst frequencies examined, we did not observe any change in protein noise with an increase in translational efficiency (*Figure 3E*). We also tested the model at different translational burst frequencies. Although changes in translational burst frequencies altered the level of protein noise, we did not find any correlation between mean protein expression and protein noise (*Figure 3F*). These results suggested that a simple model integrating transcriptional and translational bursting could not explain the observed relationship between translational efficiency and protein noise.

## Inclusion of ribosome demand reveals a positive correlation between translational efficiency and protein noise

Next, we analyzed several molecular models that could possibly explain the positive correlation between translational efficiency and protein noise. An increase in translational efficiency, leading to a higher translation rate, will require an increased supply of tRNA molecules and ribosomes. Could an increase in translation rate lead to a bottleneck in the supply of tRNA and generate variation in protein expression across cells? This seemed highly unlikely, as high translational efficiency is associated with the presence of preferred codons, for which abundant tRNA molecules are available in a cell (*dos Reis et al., 2004*; *Tuller et al., 2010*). In comparison, rare codons have lower numbers of corresponding tRNA molecules available (*dos Reis et al., 2004*; *Tuller et al., 2010*). Thus, the presence of rare codons would, in fact, cause a bottleneck in tRNA supply and hence, would lead to higher expression noise. This would further imply that a reduction in translational efficiency would generate higher expression noise, which would contradict the actual empirical observations (*Ozbudak et al., 2002*; *Blake et al., 2003*).

The translational efficiency is dependent on the availability of ribosome machinery in the cell. We analyzed whether higher translational efficiency could lead to a bottleneck in the availability of ribosomes for mRNA molecules of a gene, cause variation in the translation process of a gene among cells, and thereby generate high expression noise. It has been reported that competition for ribosomes rather than tRNA imposes constraints on translation in yeast (*Chu et al., 2011*). In addition,

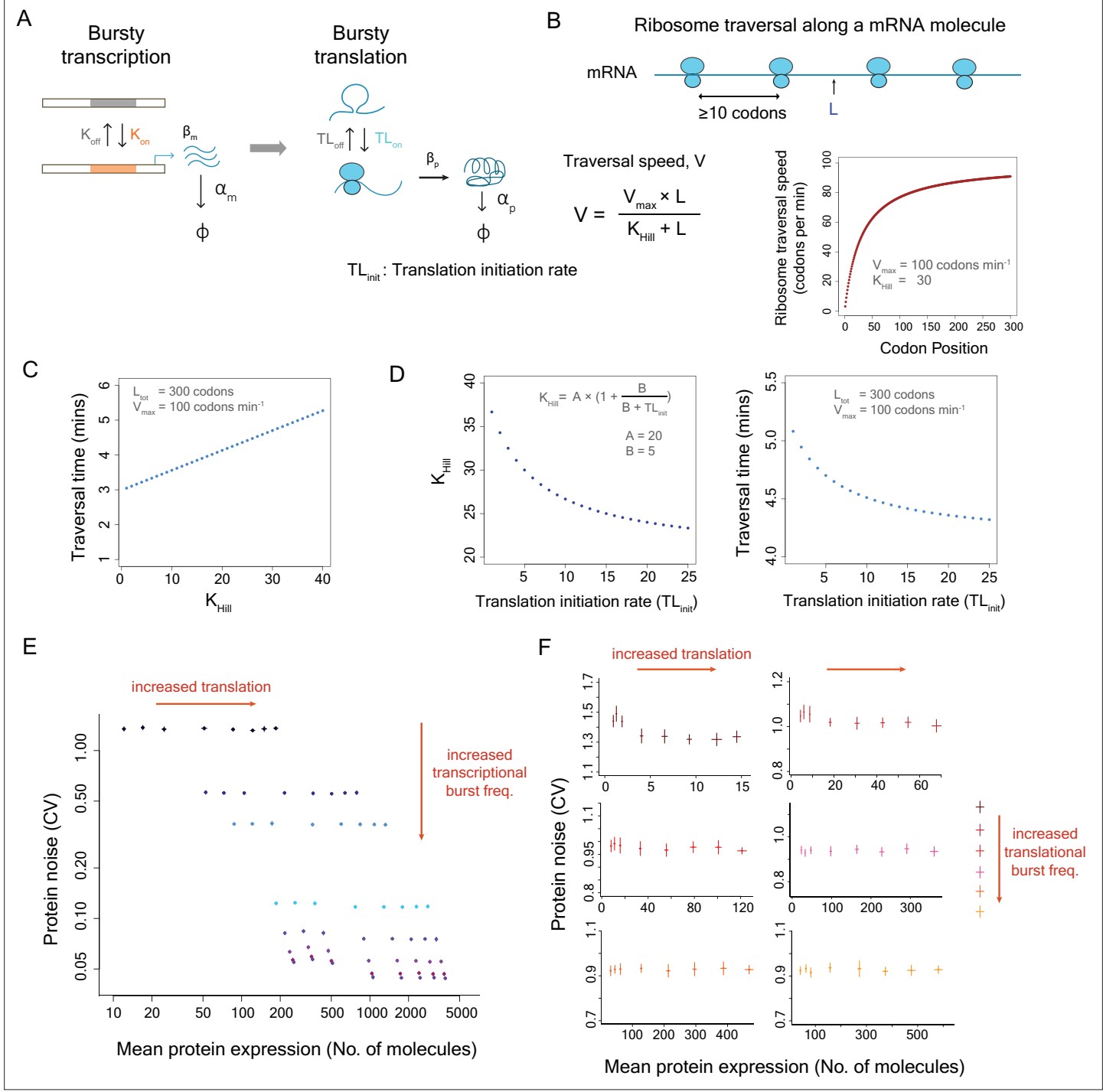

**Figure 3.** The model combining transcriptional and translational bursting does not explain the positive correlation between translational efficiency and protein noise. (**A**) Schematic diagram depicting the integrated model of transcriptional and translational bursting. (**B**) Ribosomal traversal speed along an mRNA molecule is given by V at a position L in the mRNA molecule. As multiple ribosomes could translate a single mRNA molecule at the same time, a second translation initiation happened only when the preceding ribosome traversed at least 10 codons, to account for steric interaction between ribosomes (***Steitz, 1969***; ***Ingolia et al., 2009***). The ribosome traversal speed was modeled using a Hill function as several studies have shown the presence of a gradually increasing profile of translational efficiency or ramp in the 5' end of coding regions of genes (***Tuller et al., 2010***; ***Weinberg et al., 2016***). (**C**) Traversal time calculated as a function of $K_{Hill}$ from the Hill function for a gene with 300 codons, and the maximum traversal speed of 100 codons per minute. (**D**) Relationship between $K_{Hill}$, translation initiation rate, and ribosome traversal time. Faster ribosome traversal enabled higher translation initiation rate ($TL_{init}$) (***Barrington et al., 2023***). A and B are parameters of the model. (**E**) The results obtained from stochastic simulations using the combined model of transcriptional and translational bursting. Protein noise changes with changes in translational efficiency and transcriptional

*Figure 3 continued on next page*

*Figure 3 continued*

burst frequency but does not reveal a positive correlation between mean protein expression and protein noise. (**F**) Protein noise obtained from the combined model changes with changes in translation initiation rate and translational burst frequency but does not explain the positive correlation between mean protein expression and protein noise.

studies in cell-free systems have shown that constraints on resources for transcription and translation can cause bursty expression (*Caveney et al., 2017*).

For a gene that is transcribed in bursts, there will be considerable temporal fluctuation in the number of mRNA molecules available for translation. When the mRNA copy number of that gene is high, they will require many ribosomes for translation. As the mRNA copy-number drops due to stochastic fluctuations in transcription rate, some of the ribosomes earlier involved in translation of mRNA molecules of this gene become available and can be allocated for translating mRNA molecules of other genes (*Figure 4A*). At a subsequent time point when the mRNA number of this gene rises again, it increases the demand for ribosomes (*Figure 4A*). In actively growing cells, where the demand for ribosomal machinery is high, this can constrain translation initiation rate for these mRNA molecules. As the allocation of ribosomes to mRNA molecules is likely to be a stochastic process, this can generate variation in translation rates and consequently, protein expression of that gene between cells.

There are two predictions that come out of this model. First, it predicts that a reduction in the high demand for ribosome machinery during the low- to high-mRNA copy-number transition will lower protein expression noise (*Figure 4A*). This can be achieved by lowering the translational efficiency of the gene. In this scenario, we will observe high protein noise at high translational efficiency and low noise at low translational efficiency, which will agree with the empirical observations.

Another way of minimizing the sudden demand for ribosomes is to reduce temporal fluctuations in mRNA numbers. This can be achieved at the level of transcription by employing a more uniform rate of transcription (*Figure 4B*). Thus, according to the second prediction, protein noise of genes that are expressed in a uniform rate of transcription will be minimally affected by changes in the translational efficiency (*Figure 4B*).

In addition, we explored whether co-translational protein folding, ribosome collision, translational errors, or changes in mRNA stability associated with translation rate could explain the positive association between translational efficiency and protein noise. Proteins are co-translationally folded as they exit through the ribosome tunnel after translation (*Liu, 2020*). The speed of translation might affect the efficiency of co-translational folding, and perhaps, fast translation could lower efficiency of co-translational folding. This can generate misfolded proteins that are degraded. Since this process is stochastic in nature, it can lead to increased variation in protein expression among cells within a population. However, several studies have reported the exactly opposite phenomenon. A study by *Spencer et al., 2012* showed that the use of optimal codons leads to better folding of the encoded polypeptide. Other studies have shown that fast translation can in fact help avoid misfolded states and can help in better co-translational folding, especially for misfolding-prone regions (*O'Brien et al., 2014*; *Trovato and O'Brien, 2017*).

High translational efficiency means fast traversal of ribosomes through an mRNA molecule, which would in turn increase the translation initiation rate. This can lead to a higher chance of collision between two successive ribosomes on an mRNA molecule in case of a slow-down in movement due to occurrence of non-preferred codons. Slower ribosome traversal reduces translation initiation rate and thus lowers the chance of ribosome collision even in case of a slow-down. Ribosome collision leads to degradation of mRNA, rescue of bound ribosome, and can also affect further translation initiation (*Simms et al., 2017*; *Juszkiewicz et al., 2020*; *Saito et al., 2022*). Therefore, this could be a possible reason for higher protein noise in genes that show high translational efficiency. However, in contrast to the ribosome demand model, this observation would hold for all genes with high translational efficiency, irrespective of their burst frequencies.

We further considered whether translational errors could explain the correlation between translation rate and protein expression noise. It is possible that fast translation can lead to more translational errors. This can generate proteins with wrong amino acids, which can result in protein misfolding and therefore can lead to protein degradation. Since translational errors occur at random, this process will vary from one cell to another, which can lead to higher protein expression noise. However, there are

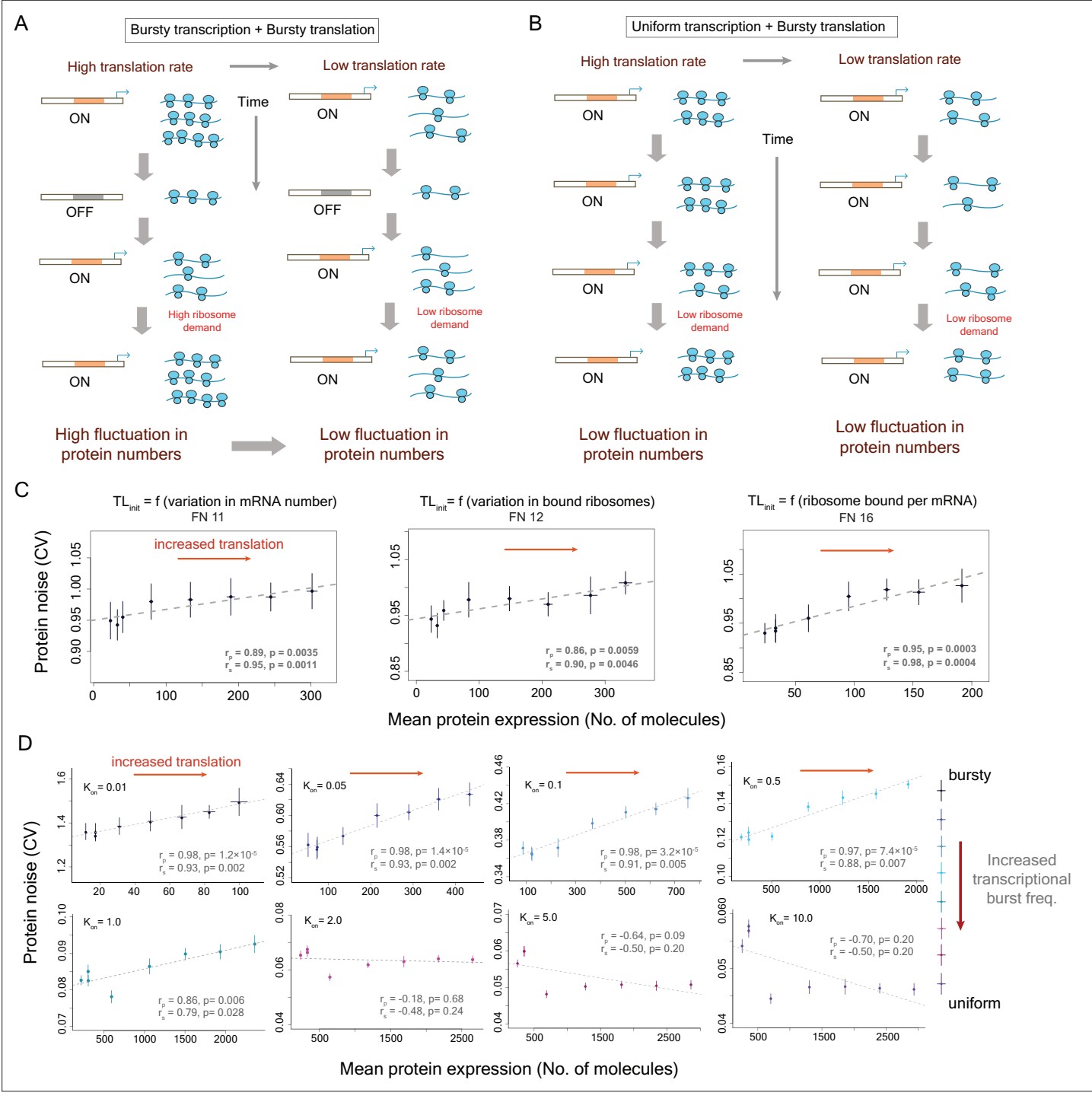

**Figure 4.** Inclusion of ribosome demand associated with translation of mRNA molecules of a gene can reveal positive correlation between translational efficiency and protein noise. (**A**) Schematic diagram depicting how ribosome demand for translation of mRNA molecules can vary with bursty transcription and bursty translation. As mRNA numbers of the gene fluctuate due to bursty transcription, high translational efficiency can lead to intermittent elevated ribosome demand for translation of the mRNA molecules of that gene. This can lead to increased inter-individual variation in protein numbers. (**B**) Uniform transcription of a gene does not lead to a sudden elevated ribosome demand for translation, thereby reducing inter-individual variation in protein numbers. (**C**) Results from simulations with three different functions (function 11, 12, and 16 from *Supplementary file 1*) that model the impact of ribosome demand on translational efficiency. Results from simulations with other functions to model ribosome demand are shown in *Figure 4—figure supplement 1*. (**D**) The relationship between mean protein expression and protein noise at different transcriptional burst frequencies obtained from stochastic simulations with the model incorporating ribosome demand along with transcriptional and translational bursting. The ribosome demand was modeled using function 16 (*Supplementary file 1*). For each transcriptional burst frequency, the translational efficiency

*Figure 4 continued*

was altered by changing the translation initiation rate (TL~init~) while keeping the rest of the parameters constant. The figures show mean ± 1 s.d. values obtained from simulations.

The online version of this article includes the following figure supplement(s) for figure 4:

**Figure supplement 1.** Relationship between mean protein expression and protein noise (coefficient of variation, CV) derived from stochastic simulations based on different mathematical functions to model ribosome demand (***Supplementary file 1***).

conflicting reports regarding this. An earlier study reported that translational errors occurred at sites where the speed of ribosome was higher (***Mordret et al., 2019***). On the other hand, a recent study observed that preferred or optimal synonymous codons, where ribosome traversal is fast, were translated more accurately (***Sun and Zhang, 2022***).

Finally, we also considered whether the translation process itself could change mRNA stability and therefore could generate cell-to-cell variation. It has been reported that increased translation can lead to higher mRNA destabilization (***Dave et al., 2023***). In addition, codon optimality and thus elongation rate has also been suggested to be positively associated with mRNA stability (***Presnyak et al., 2015***). A more recent study has, however, shown that translation initiation rate, but not elongation rate, is the major determinant of mRNA stability and inhibition of translation initiation has been shown to increase mRNA decay (***Chan et al., 2018***). However, if changes in the translation rates altered mRNA stability and led to increased variation in cell-to-cell variation in protein concentration, this will be observed for all genes irrespective of their burst characteristics.

Since the ribosome demand model agreed with the earlier empirical observations, we tested whether incorporating ribosome demand in our mathematical model could capture the observed positive correlation between translational efficiency and protein noise. The ribosome demand for translating the mRNA molecules of a specific gene was modeled as a resource allocation problem that depended on the fluctuation in the number of mRNA molecules of that gene and the number of ribosomes that were available for translation of these mRNA molecules (***Figure 4A***). Large variation in mRNA numbers led to large fluctuations in ribosome demand which also affected translation initiation rate. For example, a sudden surge in the number of mRNA molecules of the gene from one time point to another necessitated more ribosomes for translation, thereby creating increased ribosome demand for translation of that gene. Similarly, large variation in the number of ribosomes required to be bound to mRNA molecules of that gene between two consecutive time points affected ribosome allocation and influenced translation initiation rate.

Therefore, we modeled ribosome demand using various mathematical functions (***Supplementary file 1***) where we considered it to be a function of variation in mRNA number, variation in ribosome bound per mRNA molecule or a combination of both (the parameter 'av', ***Supplementary file 1***). The translation initiation rate (TL~init~) at a specific time point varied from the basal translation initiation rate according to Hill functions incorporating ribosome demand (***Supplementary file 1***). The Hill functions were of the order 10 to model a sharp decrease in translation initiation rate if the ribosome demand was too high. For each of the functions for modeling ribosome demand, we performed stochastic simulations and changed mean expression by varying the basal translation initiation rate while keeping other parameters constant (***Figure 4C***, ***Figure 4—figure supplement 1***).

Interestingly, the functions where translation initiation rate was dependent on the number of ribosomes bound per mRNA molecule showed the most positive association between mean protein expression and protein noise (***Figure 4C***, ***Figure 4—figure supplement 1***). We, therefore, considered one of these functions (function 16) for all subsequent analysis (***Figure 4C***, ***Supplementary file 1***). However, we observed similar positive associations for several variants of these functions, suggesting that the association found was robust to variations in the actual form of the function (***Supplementary file 1***, ***Figure 4—figure supplement 1***).

To further test whether our mathematical model could capture the two predictions of the ribosome demand model, we performed stochastic simulations for a wide range of transcriptional and translational burst frequencies (***Figure 4D***). At low transcriptional burst frequencies, protein noise increased with an increase in translation rate (***Figure 4D***). In addition, as we increased the transcriptional burst frequency and moved toward a more uniform rate of transcription, the translation rate had minimal effect on protein noise (***Figure 4D***). Thus, the mathematical model could recapitulate both predictions of the ribosome demand model.

We also tested the impact of translational burst frequency on the association between translational efficiency and protein noise (*Figure 5A*). At very low translational burst frequencies, protein noise did not show much change with an increase in translation rate (*Figure 5A*). This was due to a very low level of protein production at a very low translational burst frequency, as the mRNA molecule mostly resided in the inactive state. At higher translational burst frequencies, we again observed an increase in protein noise with an increase in the translation rate and hence with an increase in mean protein expression (*Figure 5A*). We further tested the robustness of our stochastic modeling approach through a systematic variation in values of several model parameters (*Figure 5*, *Figure 5—figure supplements 1–6*), as well as explored different scenarios through random sampling of the parameter space (*Figure 5—figure supplements 7–9*).

Next, we investigated whether elimination of the ribosome demand from our model could lead to a loss of the positive correlation between mean protein expression and protein noise. To do so, we tested two sets of models where we decoupled the translation initiation rate ($TL_{init}$) from the ribosome traversal speed ($V$) and changed each of these parameters independently without varying the other. In the first set of models and stochastic simulations, we altered the mean protein expression of the gene by altering the base translation initiation rate but keeping the ribosome traversal speed constant. In these models, ribosome demand varied with changes in the basal translation initiation rate and maintained the positive correlation between mean protein expression and protein noise (*Figure 5B*). The results were similar for ribosome traversal speed set at different values defined by the Hill coefficient ($K_{Hill}$) according to *Equation 10*. In the second set of models and simulations, we changed the mean protein expression by altering the ribosome traversal speed but keeping the translation initiation rate constant. These models did not allow ribosome demand to vary according to the changes in the ribosome traversal speed and showed a loss of positive correlation between mean protein expression and protein noise (*Figure 5C*, *Figure 5—figure supplement 10*). Thus, these results suggested that the ribosome demand was necessary for maintaining the positive correlation between mean protein expression and protein noise.

## Impact of coding sequence mutations on protein noise is dependent on the burst characteristics of the promoter

To empirically test the predictions of the ribosome demand model, we set up an experimental assay to measure protein expression noise in the yeast model system (*Figure 6A*). We built gene constructs where the expression of a green fluorescent protein (*GFP*) gene could be controlled by different promoters with different burst frequencies (*Figure 6—figure supplement 1*). We then introduced several mutant variants of the GFP gene with altered translational efficiencies, under the regulation of these promoters. We integrated all constructs into the yeast genome to eliminate expression noise arising from fluctuations in plasmid copy number (*Figure 6A*). We measured GFP expression using flow cytometry (*Figure 6—figure supplement 2*), and estimated noise from a homogeneous group of cells that showed very similar cell size and cell complexity (*Silander et al., 2012*; *Figure 6B*, *Figure 6—figure supplement 3*).

We picked four promoters for our constructs (*Figure 6C*) – two promoters, of *RPG1* and *RPL35A* genes, with high burst frequencies ($K_{on} = 22.855$ and $K_{on} = 47.464$, respectively) and low protein noise (DM = −0.841 and DM = −1.4, respectively) (*Newman et al., 2006*), and two promoters, of *CPA2* and *QCR2* genes, with low burst frequencies ($K_{on} = 0.704$ and $K_{on} = 1.186$, respectively) and high protein noise (DM = 9.583 and DM = 3.515, respectively) (*Newman et al., 2006*). Our measurements also confirmed that the promoters *CPA2* and *QCR2* had higher protein noise compared to *RPG1* and *RPL35A* (*Figure 6D*).

To change translational efficiency in all gene constructs in a similar manner, we built variants of the GFP gene that differed in their codon usage. This altered the translation elongation rate, which in turn would also alter the translation initiation rate (*Barrington et al., 2023*). We constructed nine mutant variants of the *GFP* gene and cloned each of them under the control of these four promoters. The variants consisted of seven single mutants and two variants with multiple mutations (see Methods). Many of the codon substitutions led to usage of nonpreferred codons in yeast. We specifically focused on mutating N- and C-terminal regions of the GFP protein so as not to affect its functional core. In addition, variants contained mostly synonymous mutations with two exceptions. We changed glutamic acids with aspartic acids at codons 5 and 234, and tyrosine with asparagine at codon 237 of the GFP

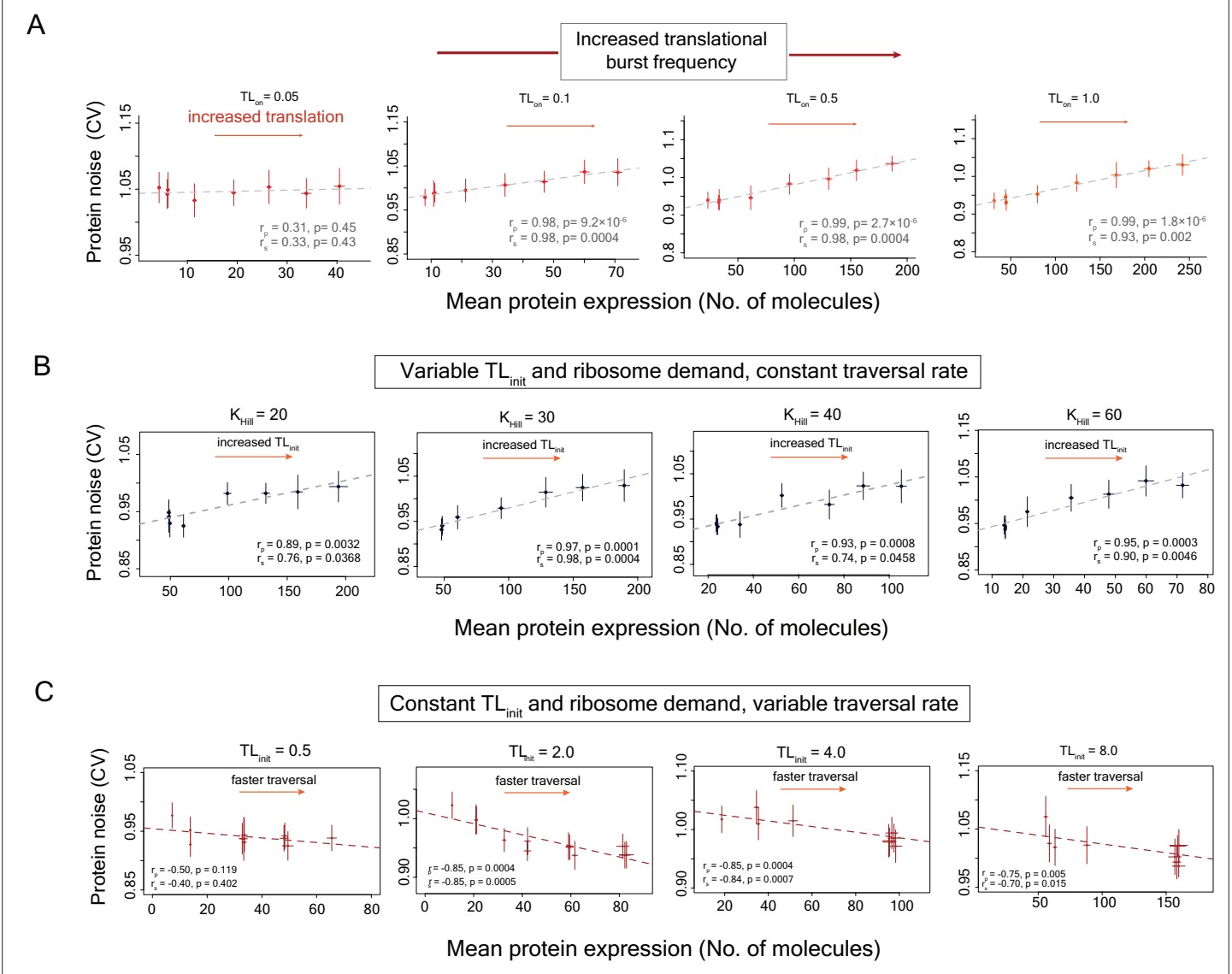

**Figure 5.** Ribosome demand is necessary for the positive correlation between mean protein expression and protein noise. (**A**) The relationship between mean protein expression and protein noise at different translational burst frequencies obtained from stochastic simulations with the model incorporating ribosome demand along with transcriptional and translational bursting. The ribosome demand was modeled using function 16 (***Supplementary file 1***). For each transcriptional burst frequency, the translational efficiency was altered by changing the translation initiation rate ($TL_{init}$) while the rest of the parameters were kept constant. (**B**) Stochastic simulations where mean protein expression was altered by changing base translation initiation rate ($TL_{init}$), thus altering ribosome demand, but keeping ribosome traversal speed constant, maintained positive correlation between mean protein expression and noise. This was done by keeping $K_{Hill}$ constant at specific values during simulations that constrained the traversal time (***Equation 10***). (**C**) Stochastic simulations where mean protein expression was altered by changing the ribosome traversal speed (***Equation 10***) but keeping the base translation initiation rate ($TL_{init}$) constant, and thus, not allowing variation in ribosome demand with changes in ribosome traversal rate, abolished the positive correlation between mean protein expression and protein noise. The figures show mean ± 1 s.d. values obtained from simulations.

The online version of this article includes the following figure supplement(s) for figure 5:

**Figure supplement 1.** Relationship between mean protein expression and protein noise (coefficient of variation, CV) derived from stochastic simulations for different values of the parameter A.

**Figure supplement 2.** Relationship between mean protein expression and protein noise (coefficient of variation, CV) derived from stochastic simulations for different values of the parameter B.

**Figure supplement 3.** Relationship between mean protein expression and protein noise (coefficient of variation, CV) derived from stochastic simulations for different values of the parameter $V_{max}$.

*Figure 5 continued on next page*

*Figure 5 continued*

**Figure supplement 4.** Relationship between mean protein expression and protein noise (coefficient of variation, CV) derived from stochastic simulations for different values of the parameter $K_{off}$.

**Figure supplement 5.** Relationship between mean protein expression and protein noise (coefficient of variation, CV) derived from stochastic simulations for different values of the parameter $\beta_m$.

**Figure supplement 6.** Relationship between mean protein expression and protein noise (coefficient of variation, CV) derived from stochastic simulations for different values of the parameter $TL_{off}$.

**Figure supplement 7.** Relationship between mean protein expression and protein noise (coefficient of variation, CV) derived from stochastic simulations for sets of parameter values obtained by random sampling of parameter space.

**Figure supplement 8.** The relationship between mean protein expression and protein noise (coefficient of variation, CV) derived from stochastic simulations for sets of parameter values obtained by random sampling of parameter space.

**Figure supplement 9.** The relationship between mean protein expression and protein noise (coefficient of variation, CV) derived from stochastic simulations for sets of parameter values obtained by random sampling of parameter space.

**Figure supplement 10.** Stochastic simulations where mean protein expression was altered by changing the ribosome traversal speed (*Equation 10*) but keeping the base translation initiation rate ($TL_{init}$) constant, and thus, not allowing variation in ribosome demand with changes in ribosome traversal speed abolished the positive correlation between mean protein expression and protein noise.

gene. We created all constructs in three replicates and estimated their noise through multiple independent experiments on different days.

The multi-mutant variant at the N-terminal end, containing five mutations (N-5), had the most number of rare codons and thus was expected to show substantial reduction in mean protein expression owing to its low translational efficiency. We compared the impact of the coding region mutations across different promoters through the measures of normalized mean expression and normalized protein noise (*Figure 6—figure supplement 4A and B*), to account for differences in mean protein expression and absolute protein noise among the four promoters. The N-terminal multi-mutant (N-5) showed substantial reduction in normalized mean protein expression (~56–81% mean expression compared to the wild-type GFP gene) across all promoters (*Figure 6—figure supplement 4A*). Very interestingly, the N-5 mutant also showed substantial reduction in protein noise values, but only when expressed under the bursty promoters *CPA2* and *QCR2* (*Figure 6—figure supplement 4B*). This mutant showed 8% and 11% reduction in normalized protein noise compared to the wild-type GFP gene when expressed under the *CPA2* and *QCR2* promoters, respectively (Mann–Whitney *U*-test, p = $1.2 \times 10^{-4}$ and p = $6.7 \times 10^{-4}$, respectively), but showed much lower change in normalized protein noise when expressed under the regulation of the promoters *RPG1* and *RPL35A* (~3% increase and ~4% decrease in CV, respectively) (*Figure 6—figure supplement 4B*), despite 36% and 19% decrease in mean expression under the regulation of these non-bursty promoters.

Analysis of the relationship between normalized protein noise and normalized mean expression across all GFP variants revealed very interesting results. The regression line fitted to the plot of normalized mean protein expression against normalized protein noise showed positive slopes only for bursty promoters *CPA2* and *QCR2* (slope of 0.229 and 0.302, respectively, *Figure 6E*), like what had been observed before (*Ozbudak et al., 2002*; *Blake et al., 2003*). In contrast, the regression line for the promoters *RPL35A* and *RPG1* showed near-zero or negative slope (slope of –0.131 and 0.0228, respectively, *Figure 6F*). Genome-wide analysis of relationship between protein noise and protein synthesis rate per mRNA or tAI for groups of genes with different transcriptional burst frequencies also showed similar trend, with stronger positive relationship between protein noise and translational efficiency for genes with bursty transcription ($K_{on} \leq Q_1$, first quartile of burst frequency) (*Figure 6—figure supplement 5*). These results were in line with the predictions from stochastic simulations, and therefore, suggested that variation in ribosome demand was indeed the molecular link between stochastic fluctuations in mRNA numbers and protein noise.

## Discussion

Taken together, our work provides unique molecular insights into the long-standing observation of positive correlation between translational efficiency and protein noise. We test the long-held assumption that low transcription rate combined with high translational efficiency can generate this positive correlation. We show that the high protein noise in genes with low transcription rate and high

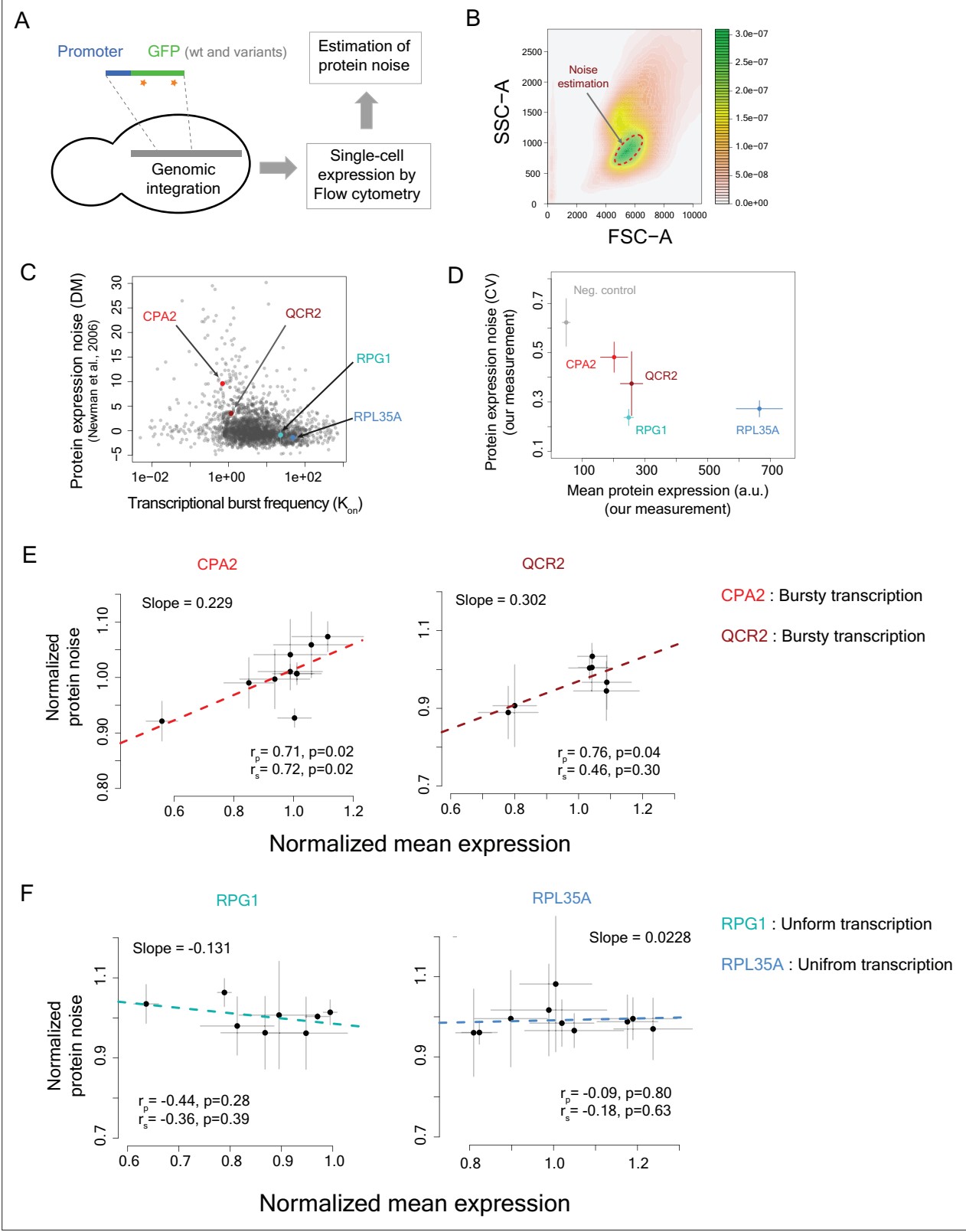

**Figure 6.** Impact of changes in translational efficiency on protein noise is dependent on the transcriptional burst characteristics of promoters. (**A**) Gene–promoter construct for genomic integration and noise measurement. (**B**) Noise estimation from a homogeneous group of cells with similar cell size and complexity. (**C**) Protein noise (distance-to-median, DM) (*Newman et al., 2006*) vs burst frequency values for yeast genes. Burst frequency values were estimated from single-cell RNA-seq data (*Nadal-Ribelles et al., 2019*) using the method described by *Kim and Marioni, 2013*. (**D**) Measured

*Figure 6 continued on next page*

*Figure 6 continued*

mean protein expression and protein noise of the promoters of *RPL35A*, *RPG1*, *CPA2*, and *QCR2* with the wild-type GFP gene. (**E**) Relationship between normalized protein noise and normalized mean expression of the GFP mutants in the bursty promoters *CPA2* and *QCR2*. (**F**) Relationship between normalized protein noise and normalized mean expression of the GFP mutants in the non-bursty promoters *RPG1* and *RPL35A*. The figures show mean ± 1 s.d. values quantified from flow cytometry experiments. Three clones for each mutant were measured.

The online version of this article includes the following figure supplement(s) for figure 6:

**Figure supplement 1.** Structure of the promoter–GFP constructs along with auxotrophic marker (*HIS3MX6*) and homologous recombination sites (His3L and His3R) for genomic integration into yeast.

**Figure supplement 2.** Snapshot of a flow cytometry experiment for measurement of mean protein expression and protein noise.

**Figure supplement 3.** Fitting of ellipses (shown by red color) with different values of major and minor axes ('a' and 'b') and identification of the best fit that chooses a homogenous set of cells and contains at least ~20,000 cells for quantification of noise.

**Figure supplement 4.** Experimental measurement of mean protein expression and protein noise for GFP mutants under the regulatory control of bursty and non-bursty promoters.

**Figure supplement 5.** Correlation between protein expression noise and protein synthesis rate and tAI for classes of genes with different busrt frequency parameter value.

**Figure supplement 6.** Construction of GFP mutant variants along with different promoters for genomic integration into yeast by overlap-extension PCR.

translational efficiency primarily stems from higher stochastic variation in mRNA numbers associated with low rate of transcription. We also show that demand for ribosomal machinery underlies translational modulation of transcriptional noise. Thus, our work reveals how the coding sequences of genes can influence expression noise, and how this contribution is dependent on the properties of the promoter. This work also highlights how dynamic properties of a system and shared resources influence gene expression, beyond genetic features such as promoter sequence or presence of specific motifs.

Our results also illustrate how the translation process can decouple mean protein expression and protein noise (*Loell et al., 2022*), with a departure from the traditional inverse relationship between them. The nature and extent of translational decoupling is dictated by the translational efficiency of the coding sequence, as well as by the promoter and its mode of transcription.

There are several other factors besides codon usage that can impact translational efficiency. For example, presence of micro-ORF upstream of the start codon and presence of mRNA secondary structures in the 5'UTR region can affect translational efficiency (*Wu et al., 2022*; *Leppek et al., 2018*). In addition, the presence of a stretch of positively charged residues in the amino acid chain can slow-down movement of the growing peptide chain through the ribosomal exit tunnel and can impact translational efficiency (*Charneski and Hurst, 2013*). Thus, the effective translational efficiency for a gene is a complex combination of multiple variables. It remains to be seen how these variables combine with translational bursts and impact expression noise.

In summary, our work identifies variation in ribosome availability as the key feature driving the translational modulation of transcriptional bursts. Such variation happens across organisms and therefore, these results have broad implications for studying protein noise across biological systems. Advancements in single-cell RNA sequencing have enabled us to study mRNA expression noise across diverse cellular processes in different organisms. However, noise at the protein level has more direct impact on cellular phenotypes and therefore, on phenotypic heterogeneity. Thus, a better understanding of how mRNA noise is linked to protein noise will help us better predict phenotypic heterogeneity. Our work further shows that combining estimates of burst parameters from single-cell RNA-sequencing data with tAI values of genes can help predict protein expression noise. As more single-cell RNA-seq data become available, this will greatly facilitate estimation of protein expression noise and will enhance our understanding of its impact on cellular processes across different biological systems.

## Materials and methods
### Datasets
The mean mRNA expression levels of yeast genes were calculated from the single-cell RNA-seq data in yeast (*Nadal-Ribelles et al., 2019*). Mean mRNA expression values for 5500 genes were obtained

from this data. Noise in mRNA expression was calculated following the method described in *Parab et al., 2022*. Briefly, for each gene, mean expression, standard deviation, and subsequently, CV values were calculated. In the next step, polynomial functions of different degrees were fitted to the CV vs log-transformed mean mRNA expression plot, and the best fit was chosen. The polynomial function of order 5 was observed to give the best fit and was used for calculating mean-adjusted mRNA expression noise. For each gene, mean-adjusted mRNA noise was calculated by the distance of the vertical line drawn from its CV value to the fitted spline. Protein expression noise values for genes were obtained from *Newman et al., 2006* and their DM values, which were corrected for dependence of CV on mean protein expression, in YPD medium were used. Protein noise of 2763 genes was obtained from this data. For translational efficiency, the data from *Riba et al., 2019* were used, and for each gene, the real protein synthesis rate per mRNA value from their data was considered as its measure of translational efficiency. The tAI values for all yeast genes were calculated using tAI value of each codon in yeast from *Sabi and Tuller, 2014*, according to the method described in *Tuller et al., 2010*.

## Two-state model for bursty transcription

Transcriptional bursts were modeled using a two-state model where a gene transitioned between on- and off-states (*Peccoud and Ycart, 1995*). The rate of transitions to on- and off-states was considered to follow Poisson distributions individually and thus, the time intervals between two on-transitions or off-transitions were exponentially distributed. The time intervals between successive events (on- or off-switching) were sampled from exponential distributions with rate parameters $K_{on}$ and $K_{off}$, respectively. Transitions to on-state resulted in production of mRNA at a rate $\beta_m$ and translation of these mRNA molecules to proteins at a rate $\beta_p$. These mRNA and protein molecules were considered to undergo removal, resulting from dilution due to cell growth and degradation, at the rates of $\alpha_m$ and $\alpha_p$, respectively.

The dynamics of transcription and translation were modeled using the following equations:

$$\textit{Change in mRNA conc.over time} : \quad \frac{d[mRNA]}{dt} = \beta_m - \alpha_m\,[\text{mRNA}], \tag{1}$$

where $\beta_m$ denoted the transcription rate per unit time (or burst size) and $\alpha_m$ denoted the removal rate of mRNA due to degradation and dilution.

Similarly,

$$\textit{Change in protein conc.over time} : \quad \frac{d[P]}{dt} = \beta_p \times [\text{mRNA}] - \alpha_p\,[\text{P}], \tag{2}$$

where $\beta_p$ denoted the protein production rate from mRNA and $\alpha_p$ denoted the protein removal rate.

Stochastic simulations were performed using Gillespie's algorithm (*Gillespie, 1977*) to model changes in the numbers of mRNA and protein molecules over time. The behavior of the system was tracked at small discrete time intervals $\Delta t$ from the initial time point $t$. These resulted in observations at '$n + 1$' time points $t$, $t + \Delta t$, $t + 2 \times \Delta t$, …, $t + n \times \Delta t$. Any event of binding or unbinding occurring within a time interval was noted and resulted in changes in transcription rate which eventually led to a change in protein concentration. As the time interval $\Delta t$ was small, the equations modeling the behavior of the systems were simplified as

$$[\text{mRNA}]_{t+\Delta t} = [\text{mRNA}]_t + \left(\beta_m - \alpha_m \times [\text{mRNA}]_t\right) \times \Delta t, \tag{3}$$

and

$$[\text{P}]_{t+\Delta t} = [\text{P}]_t + \left(\beta_p \times [\text{mRNA}]_t - \alpha_p \times [\text{P}]_t\right) \times \Delta t, \tag{4}$$

$\beta_m$, $\alpha_m$, $\beta_p$, and $\alpha_p$ were expressed in appropriate units for further simplification of these equations. The base parameter values for the model were set as: $K_{on}$ = 0.02, $K_{off}$ = 0.2, $\beta_m$ = 10 per unit time, $\beta_p$ = 100 per mRNA molecule per unit time, $\alpha_m$ = 0.07 per mRNA molecule per unit time, and $\alpha_p$ = 0.007 per protein molecule per unit time. The dynamics of transcription, variation in the mRNA concentration, and variation in protein concentration with time were modeled across 10,000 cells. Noise was

expressed as the CV from the calculation of mean and standard deviation in the protein level across these 10,000 cells across all time points.

To investigate how the mean mRNA expression level and the transcriptional burst frequencies combine with translation rate ($\beta_p$) and impact protein noise, simulations were performed over a wide range of values of the parameters $K_{on}$ and $\beta_m$. The parameter $K_{on}$ was varied between 0.001 and 10, and the parameter $\beta_m$ was varied between 1 and 1000 per unit time. For each value of these parameters, translation rate $\beta_p$ was varied between a range of 1 and 1000 per mRNA molecule per unit time.

## Estimation of burst parameters from single-cell RNA-seq data

The two-state model of gene expression enabled estimation of the parameters of transcriptional bursts from single-cell RNA-sequencing data using a maximum-likelihood approach, as described by *Kim and Marioni, 2013*.

Specifically,

$$p \mid K_{on}, K_{off} \sim \text{Beta}(K_{on}, K_{off}), \tag{5}$$

$$x \mid \beta_m, p \sim \text{Poisson}(\beta_m \cdot p), \tag{6}$$

where '$x$' is the number of RNA transcripts of a given gene within a cell and '$p$' is an auxiliary variable following a beta distribution. The mean of the parameter '$p$' is equal to the fraction of time that a gene spends in the active state. The resultant marginal distribution is known as Poisson-Beta distribution, denoted by $P(x|K_{on}, K_{off}, \beta_m)$, and gives the steady-state distribution for the mRNA copy numbers across cells. All parameters are further normalized by $\alpha_m$. $K_{on}$ and $K_{off}$ control the shape of the Beta distribution and represent the probability of a gene being in 'on' and 'off' states, respectively. $\beta_m$ is the mean of Poisson distribution and represents the average rate of gene expression when the gene is in the 'on' state.

The yeast single-cell RNA-seq data (*Nadal-Ribelles et al., 2019*) was given as the input to the model. The two-state model parameters $\theta = (K_{on}, K_{off}, \beta_m)$ were inferred by maximizing the likelihood of the parameters of Poisson-Beta distribution for a given set of observed data points $X$. This can be represented as:

$$arg_{max(\theta \, in \, \Theta)} = \prod_{\{x \, in \, X\}} P(x|\theta). \tag{7}$$

This was further represented as maximization of the log-likelihood, which was equivalent to minimization of the negative log-likelihood:

$$arg_{max(\theta \, in \, \Theta)} = -\sum_{\{x \, in \, X\}} \ln(P(x|\theta)), \tag{8}$$

$$arg_{min(\theta \, in \, \Theta)} = -\sum_{\{x \, in \, X\}} \ln(P(x|\theta)). \tag{9}$$

The negative log-likelihood was minimized using L-BFGS-B approach (*Liu and Nocedal, 1989*) and the resulting distributions for parameters in $\theta$ were calculated.

## Mathematical model of translation at single mRNA level for capturing bursty translation

Several mathematical models for the translation process have been described, and they include the TASEP model, the ribosome flow model, and their variants (*Zia et al., 2011*; *Andreev et al., 2018*; *Reuveni et al., 2011*; *Cook et al., 2009*; *Brackley et al., 2011*; *Jain et al., 2022*). Totally Asymmetric Simple Exclusion Process or TASEP model assumes that an mRNA molecule is a one-dimensional lattice through which a ribosome can hop in only one direction from one site to the next, provided the latter is empty.

To capture the dynamics of translation, a simple TASEP-based model with appropriate modifications was developed and bursty translation was incorporated. To start with, translational bursts were assumed to arise due to stochastic transitions of an mRNA molecule between active and inactive states with the rates $TL_{on}$ and $TL_{off}$, respectively. In the active state, translation initiation on the mRNA molecule occurred at a rate $TL_{init}$. The parameters $TL_{on}$ and $TL_{off}$ were assumed to follow Poisson

distributions individually and hence, the time intervals between two subsequent on-states (or off-states) were assumed to follow exponential distributions. The parameter values were chosen within the range of parameter values observed or estimated from experimental datasets, to build a realistic model. For yeast genes, maximum-likelihood estimates of $K_{on}$ and $K_{off}$ from single-cell RNA-seq data (**Nadal-Ribelles et al., 2019**) were in the ranges of 0.005–744, and 0.001–1000, respectively. In yeast, the experimental estimates for mRNA synthesis rates ranged from ~0.01 to 4 per min (**Miller et al., 2011**), and protein synthesis rates per mRNA ranged from 0 to 13.6 per s per mRNA (**Riba et al., 2019**). Earlier estimates for mRNA half-life ranged from 3 to 300 min (**Geisberg et al., 2014**), but a recent non-invasive measurement of mRNA half-life has found that the majority of mRNA molecules in yeast have half-lives of less than 10 min (**Chan et al., 2018**). The estimates for protein half-life ranged from 2 to ~16,000 min (**Belle et al., 2006**). However, no estimates for $TL_{on}$ and $TL_{off}$ were available. Large values of $K_{on}$, $TL_{on}$, mRNA synthesis rate, protein synthesis rates, mRNA half-life, and protein half-life made simulations extremely slow and computationally very demanding, as the model had to track many mRNA molecules and translation initiation events. Therefore, these values were not explored. In addition, genes with very stable mRNA molecules are unlikely to show fluctuation in mRNA numbers and therefore are not of interest to this study.

As multiple ribosomes could translate a single mRNA molecule at the same time, a second translation initiation happened only when the earlier mRNA bound ribosome had moved by 10 or more codons on the mRNA molecule, thus, accounting for steric interactions between two ribosomes (**Steitz, 1969**; **Ingolia et al., 2009**). Movement of the ribosome through an mRNA molecule depends on the individual codons that are present. Preferred codons for which tRNA molecules are readily available are traversed quickly, whereas the rare codons are read slower. (**Tuller et al., 2010**) observed a specific and conserved translational profile across several species where translational efficiency was low at the start of the genes for about 30–50 codons, followed by a gradual increase to the maximum efficiency. Similarly, **Weinberg et al., 2016** observed a slow rate of translation, although to a different degree, in the first 200 nucleotides of a gene. This enabled us to model ribosome traversal speed in a more general manner rather than focusing on the occurrence of specific codons that would vary from one gene to another. The ribosome traversal speed was modeled using a first-order Hill function of the following form:

$$V = \frac{V_{max}L}{K_{Hill} + L},$$ (10)

where $V$ = traversal speed of the ribosome at codon $L$ in the mRNA molecule, $V_{max}$ = maximum traversal speed of the ribosome in the mRNA molecule, $K_{Hill}$ = constant, and $L$ = position in the mRNA. The values of $V_{max}$ and $K_{Hill}$ were chosen to be 100 codons per min and 30, respectively. All parameters were further varied and tested for model robustness as discussed below.

The value of $K_{Hill}$ influenced the relationship between the traversal speed and the position $L$ and eventually determined the time taken for a ribosome to completely traverse through an mRNA molecule. High values of $K_{Hill}$ reduced the number of translation initiation events, thus mimicking the scenario of lower translational efficiency. This is because a ribosome took longer time to traverse through the first 10 codons, which was required before the next translation initiation event could take place. A reduction in the value of $K_{Hill}$ allowed higher translation initiation rate ($TL_{init}$) and captured the behavior of the system in case of high translational efficiency. Conversely, increasing the translation initiation rate required the $K_{Hill}$ value to be lower. Thus, the relationship between $K_{Hill}$ and translation initiation rate ($TL_{init}$) was modeled using the equation:

$$K_{Hill} = A \times \left(1 + \frac{B}{B + TL_{init}}\right),$$ (11)

where $A$ and $B$ were constants. The values of $A$ and $B$ shaped the relationship between $K_{Hill}$ and $\beta_p$, and in our model were chosen as 20 and 5, respectively. Higher translational efficiency led to faster traversal of the ribosome through an mRNA molecule and allowed more translation initiation events leading to higher value of $\beta_p$. The model was tested with different values of the parameters $K_{Hill}$, $A$, $B$, and $V_{max}$ to test for robustness.

In the next step, the model of translation at the single mRNA level was integrated with the two-state model of transcription, and the combined model was utilized for subsequent stochastic simulations.

The base parameter values for the model were chosen as: $K_{on}$ = 0.02, $K_{off}$ = 0.2, $\beta_m$ = 4 per minute, $TL_{on}$ = 0.5, $TL_{off}$ = 0.5, $A$ = 20, $B$ = 5, $K_{Hill}$ = 30, $V_{max}$ = 100 codons per minute, length of the gene ($L_{tot}$) = 300 codons, mRNA half-life ($HL_{mRNA}$) = 10 min, and protein half-life ($HL_{prot}$) = 100 min.

Every mRNA molecule generated through transcriptional burst was tracked by the translation model at every minute over the lifetime of the mRNA molecule, and the protein expression at every time point was estimated. Gillespie's algorithm was used for simulations in 1000 cells through which estimates for population-wide mean protein expression and protein noise were derived. The process was repeated for a wide range of transcriptional and translational burst frequency values, as well as for other parameters of the models, to investigate whether positive correlation between mean protein expression and protein noise could be observed at specific parameter values. The $K_{on}$ parameter was varied between 0.01 and 10, the $K_{off}$ parameter was varied between 0.01 and 10, $TL_{on}$ between 0.01 and 5, $TL_{off}$ between 0.01 and 5, and $\beta_m$ between 0.1 and 20 per min. For each of these parameter values, $TL_{init}$ was varied between 0.1 and 10 per mRNA molecule per minute. The values of mRNA and protein half-lives were stochastically varied by ±10% of the chosen value, as the half-lives of individual mRNA and protein molecules can vary within a cell.

## Modeling and simulation of demand for ribosomal machinery

As multiple mRNA molecules share a common pool of ribosomal machinery for translation, the number of ribosomes available for translating a specific mRNA molecule can vary with time. Several studies have modeled the translation process with finite resources and competition; however, none of them had investigated the impact on protein expression levels or on heterogeneity. Only one earlier study, investigating the stochastic nature of translation using mathematical models, predicted that use of rare codons will increase expression heterogeneity (*Garai et al., 2009*), which contradicted the empirical observations.

Competition for a finite pool of ribosomal machinery would impact the translation initiation rate in an mRNA molecule. High availability of ribosomal machinery would lead to high-translation initiation rate, whereas a scarcity of ribosomes would lower translation initiation rate. Transcriptional bursts generate stochastic fluctuations in mRNA numbers over time, which can lead to a variable demand for ribosomal machinery required for translation. When the number of mRNA molecules of a gene increases suddenly due to a transcriptional burst, the demand for ribosomal machinery increases rapidly. Subsequently, when the mRNA numbers fall, this lowers demand and frees up ribosomes, which can then be utilized for translating mRNA molecules of other genes.

The demand for ribosomal machinery was incorporated into the model of translation so that its effect on translation initiation rate of an mRNA molecule could be captured. For a gene, the number of mRNA molecules produced by transcriptional bursting and the number of ribosomes bound to mRNA molecules were tracked over time. At every time point during simulation, the ribosome demand was modeled using different functions that incorporated the ratio of mRNA numbers at the current time point to the mRNA numbers at the previous time point, the ratio of current number of bound ribosomes to the number of bound ribosomes at the previous time point or a combination of both (parameter 'av', *Supplementary file 1*). These included functions where the translation initiation rate at a specific time point was equal to the basal translation initiation rate divided by the ribosome demand, or more complex Hill functions of order 10 (*Supplementary file 1*), that could induce sharp transitions in translation initiation rate when demand was too high. The parameter 'av' modeled the ribosome demand at a specific time point during the simulations, and the parameter 'kc' defined the threshold value for ribosome demand beyond which the translation initiation rate decreased sharply.

A higher number of mRNA molecules or number of ribosomes bound to mRNA at a time point compared to the previous time point meant lower availability of ribosomal machinery for translation and lower translation initiation rate. Among all functions tested, the one which considered both number of mRNA molecules and number of bound ribosomes at current and previous time points, and modeled the impact of ribosome availability on translation initiation rate through Hill function (functions 14–19, *Supplementary file 1*) generated the best positive correlation between mean protein expression and protein noise (*Figure 4—figure supplement 1*). The function 16 was used for all subsequent analysis.

The integrated model of bursty transcription and bursty translation incorporating ribosome demand was tested in a wide range of values of several parameters. The impact of variation in values of the

parameters $A$, $B$, and maximum ribosome traversal rate ($V_{max}$) individually on correlation between protein noise and translational efficiency was tested. The parameter $A$ was varied between 10 and 50, $B$ between 1 and 10, and $V_{max}$ between 50 and 150. For each of the parameter values, the translation initiation rate ($TL_{init}$) was varied between 0.1 and 10 per mRNA molecule per min. For $V_{max} = 50$, no protein expression was observed. For the rest of the parameter values, no effect on correlation between translational efficiency and protein noise was observed (*Figure 5—figure supplements 1–3*).

In the next step, the model was tested with different values of the parameters $K_{on}$, $K_{off}$, $TL_{on}$, $TL_{off}$, and $\beta_m$. Each of these parameters was varied individually and the impact of changes in the translation initiation rate ($TL_{init}$) on protein noise was tested (*Figure 4D, E*; *Figure 5—figure supplements 4–6*). The parameter $K_{on}$ was varied between 0.01 and 10, $K_{off}$ between 0.01 and 10, $TL_{on}$ between 0.01 and 5, $TL_{off}$ between 0.01 and 5, and $\beta_m$ between 0.1 and 20. For each of these parameter values, $TL_{init}$ was varied between 0.1 and 10 per mRNA molecule per min. Higher rates of $\beta_m$ and $TL_{init}$ made simulations extremely slow and computationally very demanding, as the model had to track a large number of mRNA molecules and translation initiation events, respectively. Therefore, these values could not be used for simulation.

Since it was not possible to explore all possible combinations of parameter values due to an extremely large number of combinations, random sampling of parameter values for $K_{on}$, $K_{off}$, $TL_{on}$, $TL_{off}$, $\beta_m$, $HL_{mRNA}$, and $HL_{protein}$ was performed from a specified range of each parameter (*Figure 5—figure supplements 7–9*). This also enabled us to study how different combinations of values of these parameters impact protein noise, and consequently, the relationship between translational efficiency and protein noise. Fifty random sampling of parameter values was done, and for each set of parameter values, the translation rate ($\beta_p$) was varied between 0.2 and 10 per mRNA molecule per min. The parameter $K_{on}$ was sampled in the range of 0.01–2, $K_{off}$ between 0.01 and 2, $TL_{on}$ between 0.1 and 5, $\beta_m$ between 0.1 and 10 per min, $HL_{mRNA}$ between 1 and 20 min, and $HL_{prot}$ between 1 and 200 min. Simulation results indicated interactions between different parameter values that also impacted the relationship between translational efficiency and protein noise (*Figure 5—figure supplements 7–9*).

## Construction of GFP–promoter fragment and genomic integration in yeast

*S. cerevisiae* BY4741 strain was used for experiments. *GFP* gene integrated into the yeast genome was used as the model system for noise quantification. The gene–promoter construct was made as follows. The *GFP* gene was amplified from the yeast strain where the *PDR5* gene was tagged with *GFP*, and the first two codons ATG and TCT were added with the help of primers (*Supplementary file 2*). For PCR amplification, genomic DNA from yeast was isolated by lithium acetate-SDS method, and 2 µl genomic DNA was used as template. Amplification using Q5 DNA polymerase and gene-specific primers was performed, and the PCR product was subsequently purified using QIAquick PCR Purification Kit (QIAGEN).

The strain BY4741 has a truncated and non-functional *HIS3* gene (*his3Δ1*). Therefore, the gene–promoter construct was inserted into the site of the *HIS3* gene through homologous recombination, and *HIS3MX6* cassette was used as the auxotrophic marker for selection of transformed colonies. The *HIS3MX6* cassette was amplified from yeast strain with *PDR5-GFP* fusion. For genomic integration, the genomic regions upstream and downstream of the *HIS3* gene, denoted by His3L and His3R, respectively (*Figure 6—figure supplement 1*), were amplified and were separately cloned into pUC19 vector with restriction digestion and ligation. The constructs were transformed into chemically competent *E. coli* cells (*Nishimura et al., 1990*), and the transformed bacterial colonies were selected on LB plates supplemented with 100 µg/ml ampicillin. The promoters, the *GFP* gene, and the *HIS3MX6* cassette were subsequently cloned step-by-step through restriction digestion with appropriate enzymes (*Figure 6—figure supplement 1*), followed by ligation with T4 DNA ligase. The constructs were transformed into *E. coli* and were selected on LB plates supplemented with ampicillin. The cloned DNA fragments were verified by restriction digestion of the constructed fragment, as well as through Sanger sequencing. Promoters of four genes, namely, *RPG1*, *RPL35A*, *CPA2*, and *QCR2* were cloned upstream of the *GFP* gene (*Supplementary file 3*). The genes *CPA2* and *QCR2* showed high protein noise, and the genes *RPG1* and *RPL35A* showed low protein noise in genome-wide noise data of yeast (*Newman et al., 2006*). For cloning the promoter sequence of a gene, the region between the start

codon of the gene and the stop codon of the previous gene on the same DNA strand was considered (**Supplementary file 3**).

In the next step, the full construct was amplified using His3L forward and His3R reverse primers (**Supplementary file 2**) using Q5 polymerase, and 20 µl of the PCR product was used for transformation into competent yeast cells following the protocol of **Gietz and Schiestl, 2007**. Briefly, yeast cells were first grown to saturation in YPD medium at 30°C overnight, and then were diluted into fresh medium, where they were allowed to grow for 4 hr. Cells were precipitated and were washed once with 0.1 M lithium acetate (LiAc) solution. Cells were resuspended in 50 µl of 0.1 M LiAc and then 5 µl salmon sperm carrier DNA along with 20 µl of PCR product were added. Further, 300 µl of PLI solution, comprising 10% (vol/vol) 1 M LiAC, 10% (vol/vol) water, and 80% (vol/vol) PEG 3350 (50% wt/vol), was added to each tube. Cells were subjected to heat shock at 42°C for 40 min, following which PLI solution was removed, and the cells were plated on synthetic complete medium supplemented with 2% glucose but without any histidine (SD-His). Colonies with genomic integration were confirmed by colony PCR and subsequently verified by Sanger sequencing.

## Generation of single- and multi-mutant variants of GFP

Seven mutant variants of the *GFP* gene were constructed, out of which three variants contained mutations in the N-terminal region and four variants contained mutations in the C-terminal region. Variants with mutations in the N- or C-terminal regions were targeted to prevent disturbing the catalytic core of the GFP protein. The mutant variants were cloned downstream of the four promoters.

The N-terminal single mutant variants consisted of TCT to TCG (S to S) mutation at codon 2, GGA to GGT (G to G) mutation at codon 4, GAA to GAG (E to E) mutation at codon 5, and GAA to GAC (E to D) mutation at codon 6. The C-terminal single mutants consisted of GAT to GAC (D to D) mutation at codon 234, CTA to CTC (L to L) mutation at codon 236, and AAA to AAG (K to K) mutation at codon 238. The multi-mutant variant at the N-terminal end (denoted by N-5) contained five mutations, namely, TCT to TCG (S to S) at codon 2, AAA to AAG (K to K) at codon 3, GGA to GGT (G to G) at codon 4, GAA to GAG (E to E) at codon 5, and GAA to GAC (E to D) at codon 6. The multi-mutant variant at the C-terminal end (denoted by C-5) contained five mutations, namely, GAT to GAC (D to D) at codon 234, GAA to GAC (E to D) mutation at codon 235, CTA to CTC mutation (L to L) at codon 236, TAC to AAC mutation (Y to N) at codon 237, and AAA to AAG mutation (K to K) at codon 238.

To construct and clone the mutant variants of the GFP gene downstream of the promoter regions, an overlap extension PCR method following the protocol described in **Higuchi et al., 1988** was used. For generating the mutant variants, primers were designed keeping the target mutation at the center with 15 bases flanking them on each side. In the first amplification step, two fragments were generated. The first fragment was from the 5′ end of the Hi3L region till the targeted mutation site in the *GFP* gene, and the second fragment was from the targeted mutation site in the *GFP* gene till the 3′ end region of His3R (**Figure 6—figure supplement 6**). Amplification was carried out using high-fidelity Q5 DNA polymerase. The reaction mixture comprised of 2.5 µl of each primer (10 µM), 10 µl of 5x Q5 reaction buffer, 1 µl of 10 mM dNTPs, 0.5 µl of Q5 DNA polymerase (New England Biolabs) and molecular biology grade water to a total volume of 50 µl. The PCR program consisted of incubation at 98°C for 1 min (initial denaturation) followed by 30 cycles of 98°C for 10 s (denaturation), 65°C for 30 s (annealing), 72°C for 1 min 30 s (extension), and a final extension at 72°C for 10 min.

The products after amplification were treated with 1 µl of ExoSAP-IT PCR product cleanup reagent (Thermo Fisher Scientific) and 0.5 µl DpnI (New England Biolabs) for 1 hr at 37°C, followed by deactivation for 20 min at 80°C. The products were then purified using QIAquick PCR Purification Kit (QIAGEN) and quantified using Qubit 4 Fluorometer (Thermo Fisher Scientific).

As the molecular weight of the amplified fragments was variable, equimolar concentration of the fragments from the previous amplification step was used in the next step. For each reaction, 0.5–1 pm of each purified fragment was used as templates. For assembly, the reaction mixture comprised of equimolar content of two template fragments, 10 µl of 5x Q5 reaction buffer, 1 µl of 10 mM dNTPs, 0.5 µl of Q5 DNA polymerase (New England Biolabs) and molecular biology grade water to a total volume of 45 µl, without any primer. The program consisted of incubation at 98°C for 1 min (initial denaturation) followed by 10 cycles of 98°C for 10 s (denaturation), 60°C for 30 s (annealing), and 72°C for 1 min 30 s (extension). On completion of 10th cycle, 2.5 µl of forward and reverse primers

(10 μM stock of each) were added, and the program was run for the next 25 cycles by raising the annealing temperature to 72°C. The reaction was terminated by a final extension at 72°C for 10 min.

The assembled mutant constructs were then directly transformed into competent yeast cells following the LiAC method as described above, and the transformants were selected on SD-His plates. The clones were confirmed by colony PCR and Sanger sequencing.

## Measurement of expression noise using flow cytometry

Three clones of wild-type GFP gene and each mutant variant were picked for noise measurement in flow cytometry. Yeast cultures were grown in SCD medium (0.67% YNB + 0.079% complete synthetic supplement + 2% glucose) at 30°C with shaking at 220 rpm. Yeast cells from glycerol stock were inoculated in SCD medium, were grown for 24 hr, and were then diluted 1:100 in fresh medium. This process was repeated one more time. The cells were then diluted 1:50 in fresh SCD and were grown for 4 hr. In the next step, cells were centrifuged and washed twice with 1X PBS (phosphate-buffered saline) buffer and were finally resuspended in 1X PBS buffer for flow cytometry. In flow cytometry, the data were acquired for 2,00,000 cells for each sample in a BD LSR Fortessa cell analyzer (*Figure 6—figure supplement 2*). To minimize heterogeneity in signal due to heterogeneity in cell size and complexity, a small homogeneous subset of cells with similar size and complexity parameters (FSC-A and SSC-A) was filtered. Ellipses with different lengths of major and minor axes were fitted to the FSC-A vs SSC-A plot, and the best fit that covered the most of homogeneous cells was chosen (*Figure 6—figure supplement 3*). This also filtered out cell aggregates or budding cells which can have higher GFP signal than single cells. The data from these cells were then used for calculation of expression noise. The data were analyzed using custom R codes based on codes from *Silander et al., 2012*. Normalized mean expression and normalized noise value for a strain in an experiment were calculated by dividing the mean expression and noise values by the corresponding values for the wild-type variant in that experiment.

## Materials availability

All yeast strains described in the manuscript are available on email request to the corresponding author.

## Code availability

All codes for data analysis and models are available in GitHub: https://github.com/riddhimandhar/TranslationModNoise, copy archived at *Dhar, 2026*.

## Acknowledgements

The authors are thankful to the members of the lab of Dr. Gayatri Mukherjee, School of Medical Science and Technology, IIT Kharagpur for help with flow cytometry.

---

# Additional information

## Funding

| Funder | Grant reference number | Author |
|---|---|---|
| Indian Institute of Technology Kharagpur | ISIRD | Riddhiman Dhar |
| Science and Engineering Research Board | ECR/2017/002328 | Riddhiman Dhar |

The funders had no role in study design, data collection, and interpretation, or the decision to submit the work for publication.

## Author contributions

Sampriti Pal, Formal analysis, Validation, Investigation, Methodology, Writing – original draft, Writing – review and editing; Upasana Ray, Formal analysis, Investigation, Writing – review and editing;

Riddhiman Dhar, Conceptualization, Data curation, Formal analysis, Supervision, Funding acquisition, Investigation, Writing – original draft, Project administration, Writing – review and editing

**Author ORCIDs**
Riddhiman Dhar ⬤ https://orcid.org/0000-0003-4642-0492

Reviewer #1 (Public review): https://doi.org/10.7554/eLife.99322.3.sa1
Reviewer #2 (Public review): https://doi.org/10.7554/eLife.99322.3.sa2
Author response https://doi.org/10.7554/eLife.99322.3.sa3

## Additional files

### Supplementary files

Supplementary file 1. List of mathematical functions explored to model ribosome demand.

Supplementary file 2. List of primers used.

Supplementary file 3. Fragment size and nucleotide sequence of the parts of the promoter–GFP constructs.

MDAR checklist

### Data availability

Flow cytometry data generated as part of this project are available in github: https://github.com/riddhimandhar/TranslationModNoise, copy archived at *Dhar, 2026* and dryad: https://doi.org/10.5061/dryad.tb2rbp06x. Single-cell RNA-seq data in yeast was obtained from *Nadal-Ribelles et al., 2019*. Data on protein synthesis rate per mRNA from *Riba et al., 2019*. Protein expression noise data was obtained from *Newman et al., 2006*.

The following dataset was generated:

| Author(s) | Year | Dataset title | Dataset URL | Database and Identifier |
| --- | --- | --- | --- | --- |
| Riddhiman D, Sampriti P, Upasana R | 2025 | Ribosome demand links transcriptional bursts to protein expression noise | https://doi.org/10.5061/dryad.tb2rbp06x | Dryad Digital Repository, 10.5061/dryad.tb2rbp06x |

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
