## [Editor Report · eLife Assessment]

This study focuses on a previously reported positive correlation between translational efficiency and protein noise. Using mathematical modeling and analysis of experimental data the authors reach the **valuable** conclusion that this phenomenon arises due to ribosomal demand. While some aspects of the work appear to be **incomplete**, the results have the potential to be of value and interest to the field of gene expression.

---

## [Referee Report · Reviewer #1 (Public review)]

Summary:

The authors use analysis of existing data, mathematical modelling and new experiments to explore the relationship between protein expression noise, translation efficiency and transcriptional bursting.

Strengths:

The analysis of the old data and the new data presented is interesting and mostly convincing.

Weaknesses:

My main concern is the analysis presented in Figure 4. This is the core of mechanistic analysis that suggests ribosomal demand can explain the observed phenomenon. Revisions have improved clarity but I am both confused by the assumptions used here in the mathematical modelling of this section. I said before, the authors assumption that the fluctuations of a single gene mRNA levels will significantly affect ribosome demand is puzzling. The author's seem to dismiss this and maybe I am missing something. However, the specific forms used in equations of table S1 seem very phenomenological and I am not sure how these can be taken as good approximations for modelling ribosome demand. Why kc has this specific form, why such a sharp hill number is appropriate. how many total ribosomes per mRNA is assumed here (if this assumption is indeed needed). Again, my intuition is that on average the total level of mRNA across all genes would stay constant and therefore there are not big fluctuations in the ribosome demand due to the burstiness of transcription of individual genes (as this on average is compensated with drop in level of other transcripts). Should not one be considering all transcripts and total ribosomes to be able to model ribosome demand?

---

## [Referee Report · Reviewer #2 (Public review)]

This work by Pal et al. studied the relationship between protein expression noise and translational efficiency. They proposed a model based on ribosome demand to explain the positive correlation between them, which is new as far as I realize. Nevertheless, I found the evidence of the main idea that it is the ribosome demand generating this correlation is weak. Below are my major and minor comments.

Major comments:

(1) Besides a hypothetical numerical model, I did not find any direct experimental evidence supporting the ribosome demand model. Therefore, I think the main conclusions of this work are a bit overstated.

(2) I found that the enhancement of protein noise due to high translational efficiency is quite mild, as shown in Figure 6A-B, which makes the biological significance of this effect unclear.

(3) The captions for most of the figures are short and do not provide much explanation, making the figures difficult to read.

(4) It would be helpful if the authors could define the meanings of noise (e.g., coefficient of variation?) and translational efficiency in the very beginning to avoid any confusion. It is also unclear to me whether the noise from the experimental data is defined according to protein numbers or concentrations, which is presumably important since budding yeasts are growing cells.

(5) The conclusions from Figure 1D and 1E are not new. For example, the constant protein noise as a function of mean protein expression is a known result of the two-state model of gene expression, e.g., see Eq. (4) in Paulsson, Physics of Life Reviews 2005.

(6) In Figure 4C-D, it is unclear to me how the authors changed the mean protein expression if the translation initiation rate is a function of variation in mRNA number and other random variables.

(7) If I understand correctly, the authors somehow changed the translation initiation rate to change the mean protein expression in Figure 4C-D. However, the authors changed the protein sequences in the experimental data of Figure 6. I am not sure if the comparison between simulations and experimental data is appropriate.

Comments on revisions:

Updated Review: The authors have satisfactorily answered all of my questions and comments. The current manuscript is much clearer and stronger than the previous one. I do not have any other questions.

---

## [Author Response]

The following is the authors’ response to the original reviews.

**Reviewer #1 (Public review):**
Summary:The authors use analysis of existing data, mathematical modelling, and new experiments, to explore the relationship between protein expression noise, translation efficiency, and transcriptional bursting.Strengths:The analysis of the old data and the new data presented is interesting and mostly convincing.

Thank you for the constructive suggestions and comments. We address the individual comments below.

Weaknesses:(1) My main concern is the analysis presented in Figure 4. This is the core of mechanistic analysis that suggests ribosomal demand can explain the observed phenomenon. I am both confused by the assumptions used here and the details of the mathematical modelling used in this section. Firstly, the authors' assumption that the fluctuations of a single gene mRNA levels will significantly affect ribosome demand is puzzling. On average the total level of mRNA across all genes would stay very constant and therefore there are no big fluctuations in the ribosome demand due to the burstiness of transcription of individual genes. Secondly, the analysis uses 19 mathematical functions that are in Table S1, but there are not really enough details for me to understand how this is used, are these included in a TASEP simulation? In what way are mRNA-prev and mRNA-curr used? What is the mechanistic meaning of different terms and exponents? As the authors use this analysis to argue ribosomal demand is at play, I would like this section to be very much clarified.

Thank you for raising two important points. Regarding the first point, we agree that the overall ribosome demand in a cell will remain mostly the same even with fluctuations in mRNA levels of a few genes. However, what we refer to in the manuscript is the demand for ribosomes for translating mRNA molecules of a single gene. This demand will vary with the changes in the number of mRNA molecules of that gene. When the mRNA copy number of the gene is low, the number of ribosomes required for translation is low. At a subsequent timepoint when the mRNA number of the same gene goes up rapidly due to transcriptional bursting, the number of ribosomes required would also increase rapidly. This would increase ribosome demand. The process of allocation of ribosomes for translation of these mRNA molecules will vary between cells, and this process can lead to increased expression variation of that gene among cells. We have now rephrased the section between the lines 321 and 331 to clarify this point.

Regarding the second point, each of the 19 mathematical functions was individually tested in the TASEP model and stochastic simulation. The parameters ‘mRNA-curr’ and ‘mRNA-prev’ are the mRNA copy numbers at the present time point and the previous time point in the stochastic simulations, respectively. These numbers were calculated from the rate of production of mRNA, which is influenced by the transcriptional burst frequency and the burst size, as well as the rate of mRNA removal. We have now incorporated more details about the modelling part along with explanation for parameters and terms in the revised manuscript (lines 390 to 411; lines 795 to lines 807).

(2) Overall, the paper is very long and as there are analytical expressions for protein noise (e.g. see Paulsson Nature 2004), some of these results do not need to rely on Gillespie simulations. Protein CV (noise) can be written as three terms representing protein noise contribution, mRNA expression contribution, and bursty transcription contribution. For example, the results in panel 1 are fully consistent with the parameter regime, protein noise is negligible compared to transcriptional noise.

Thank you for referring to the paper on analytical expressions for protein noise. We introduced translational bursting and ribosome demand in our model, and these are linked to stochastic fluctuations in mRNA and ribosome numbers. In addition, our model couples transcriptional bursting with translational bursting and ribosome demand. Since these processes are all stochastic in nature, we felt that the stochastic simulation would be able to better capture the fluctuations in mRNA and protein expression levels originating from these processes. For consistency, we used stochastic simulations throughout even when the coupling between transcription and translation were not considered.

**Reviewer #1 (Recommendations for the authors):**
(1) Figure 1B shows noise as Distance to Median (DM) that can be positive or negative. It is therefore misleading that the authors say there is a 10-fold increase in noise (this would be relevant if the quantity was strictly positive). How is the 10-fold estimated? Similar comments apply to Figure 1F and the estimated 37-fold. I also wonder if the datasets combined from different studies are necessarily compatible.

We have now changed the statements and mentioned the actual noise values for different classes of genes rather than the fold-changes (lines 111-113 and 143-145). We agree that the measurements for mRNA expression levels, protein synthesis rates and protein noise were obtained from experiments done by different research labs, and this could introduce more variation in the data. However, it is unlikely the experimental variations are likely to be random and do not bias any specific class of genes (in Fig. 1B and Fig. 1F) more than others.

(2) How Figure 1D has been generated seems confusing, the authors state this is based on the Gillespie algorithm, but in panel 1C and also in the methods, they are writing ODEs and Equations 3 and 4 stating the Euler method for the solution of ODEs. Also, I am concerned if this has been done at steady-state. The protein noise for the two-state model can be analytically obtained, and instead of simulations, the authors could have just used the expression. Also, Figure 1D shows CV while the corresponding data Figure 1B is showing mean adjusted DM. So, I am not sure if the comparison is valid. I am also very confused about the fact that the authors show CV does not depend on the mean expression of proteins and mRNA. Analytical solutions suggested there is always an inverse relationship exists between CV and mean and this has also been experimentally observed (see for example Newman et al 2006).

We used Gillespie algorithm for stochastic simulations and identified the time points when an event (for example, switching to ON or OFF states during transcriptional bursting) occurred. If an event occurred at a time point, the rates of the reactions were guided by the equations 3 and 4, as the rates of reactions were dependent on the number of mRNA (or protein) molecules present, production rates and removal rates.

For all published datasets where we had measurements from many genes/promoters, we used the measures of adjusted noise (for mRNA noise) and Distance-to-median (DM, for protein noise). These measures of noise are corrected mean-dependence of expression noise (Newman *et al*., 2006). For simulations, which we performed for a single gene, and for experiments that we performed on a limited number of promoters, we used the measure of coefficient of variation (CV) to quantify noise, as calculation of adjusted noise or DM was not possible for a single gene.

The work of Newman *et al.* (2006) measures noise values of different genes with different transcriptional burst characteristics and different mRNA and protein removal rates. We also see similar results in our simulations (Fig. 1E), where as we increase the mean expression by changing the transcriptional burst frequency, the protein noise goes down.

(3) Estimating parameters of gene expression using reference 44 ignores the effect of variability in capture efficiency and cell size. In a recent paper, Tang et al Bioinformatics 39 (7), btad395 2023 addressed this issue.

Thank you for referring to the work of Tang et al. (2023). We note that the cell size and capture efficiency have a small effect on the burst frequency (Kon) but has a more pronounced effect on burst size (Tang et al., 2023). In our analysis, we considered only burst frequency and even with likely small inaccuracies in our estimation of Kon, we can capture interesting association of burst frequency with noise trends.

(4) In the methods "αp = 0.007 per mRNA molecule per unit time", I believe it should be per protein molecule per unit time.

Corrected.

(5) Figure 3 uses TASEP modelling but the details of this modelling are not described well.

We have now expanded the description of the modelling approach in the revised manuscript (lines 391-412; lines 693-776 and lines 797-809). In addition, we have also added more details in the figure captions.

(6) Another overall issue is that when the authors talk about changes in burst frequency or changes in translation efficiency, it is not always clear, is this done while keeping all the other parameters constant therefore changing mean expressions, or is this done by keeping the mean expressions constant?

To test for the association between mean protein expression and protein noise, we have varied the mean expression by changing the translation initiation rate (TLinit) for the most part of the manuscript while keeping other parameters constant. In figure 5, where we decoupled TLinit from ribosome traversal rate (V), we changed the mean protein expression by changing the ribosome traversal rate while keeping other parameters constant. We have now mentioned this in the manuscript.

(7) I believe Figures 5 and 6 present the same data in different ways, I wonder if these can be combined or if some aspect of the data in Figure 5 could go to supplementary. Also, the statistical tests in Figure 5E and F are not clear what they are testing.

We have now moved figures 5E and 5F to the supplement (Fig. S20). We have also added details of the statistical test in the figure caption.

**Reviewer #2 (Public review):**
This work by Pal et al. studied the relationship between protein expression noise and translational efficiency. They proposed a model based on ribosome demand to explain the positive correlation between them, which is new as far as I realize. Nevertheless, I found the evidence of the main idea that it is the ribosome demand generating this correlation is weak. Below are my major and minor comments.

Thank you for your helpful suggestions and comments. We note that the direct experimental support required for the ribosome demand model would need experimental setups that are beyond the currently available methodologies. We address the individual comments below.

Major comments:(1) Besides a hypothetical numerical model, I did not find any direct experimental evidence supporting the ribosome demand model. Therefore, I think the main conclusions of this work are a bit overstated.

Direct experimental evidence of the hypothesis would require generation of ribosome occupancy maps of mRNA molecules of specific genes at the level of single cells and at time intervals that closely match the burst frequency of the genes. This is beyond the currently available methodologies. However, there are other evidences that support our model. For example, earlier work in cell-free systems have showed that constraining cellular resources required for transcription or translation can increase expression heterogeneity (Caveney *et al*., 2017). In addition, the ribosome demand model had two predictions both of which could be validated through modelling as well as from our experiments.

To further investigate whether removing ribosome demand from our model could eliminate the positive mean-noise correlation for a gene, we have now tested two additional sets of models where we decoupled the translation initiation rate (TLinit) from the ribosome traversal speed (V). In the first model, we changed the mean protein expression by changing the translation initiation rate but keeping the ribosome traversal speed constant. Thus, in this scenario, ribosome demand varied according to the variation in the translation initiation rate. As expected, the positive correlation between mean expression and protein noise was maintained in this condition (Fig. 5B). In the second model, we changed the mean expression by changing the ribosome traversal speed but keeping the translation initiation rate (and therefore, the ribosome demand) constant. In this situation, the relationship between mean expression and protein noise turned negative (Fig. 5B and fig. S16). These results further pointed that the ribosome demand was indeed driving the positive relationship between mean expression and protein noise.

(2) I found that the enhancement of protein noise due to high translational efficiency is quite mild, as shown in Figure 6A-B, which makes the biological significance of this effect unclear.

We agree with the reviewer’s comment that the effect of translational efficiency on protein noise may not be as substantial as the effect of transcriptional bursting, but it has been observed in studies across bacteria, yeast, and *Arabidopsis* (Ozbudak *et al*., 2003; Blake *et al.,* 2003; Wu *et al.*, 2022). In addition, the relationship between translational efficiency and protein noise is in contrast with the inverse relationship observed between mean expression and noise (Newman *et al.*, 2006; Silander *et al*., 2012). We also note that the goal of the manuscript was not to evaluate the relative strength of these associations, but to understand the molecular basis of the influence of translational efficiency on protein noise.

(3) The captions for most of the figures are short and do not provide much explanation, making the figures difficult to read.

We have revised the figure captions to include more details as per the reviewer’s suggestion.

(4) It would be helpful if the authors could define the meanings of noise (e.g., coefficient of variation?) and translational efficiency in the very beginning to avoid any confusion. It is also unclear to me whether the noise from the experimental data is defined according to protein numbers or concentrations, which is presumably important since budding yeasts are growing cells.

For all published datasets where we had measurements from many genes/promoters, we used the measures of adjusted noise (for mRNA noise) and Distance-tomedian (DM, for protein noise). These measures of noise are corrected mean-dependence of expression noise. For simulations, which we performed for a single gene, and for experiments that we performed on a limited number of promoters, we used the measure of coefficient of variation (CV) to quantify noise, as calculation of adjusted noise or DM was not possible for a single gene. We now mention this in line 123-124. We used the measure of protein synthesis rate per mRNA as the measure of translational efficiency (Riba *et al*., 2019; line 100). Alternatively, we also used tRNA adaptation index (tAI) as a measure of translational efficiency, as codon choice could also influence the translation rate per mRNA molecule (Tuller *et al*., 2010) (line 193).

The protein noise was quantified from the signal intensity of GFP tagged proteins (Newman *et al.,* 2006; and our data), which was proportional to protein numbers without considering cell volume. For quantification of noise at the mRNA level, single-cell RNA-seq data was used, which provided mRNA numbers in individual cells.

(5) The conclusions from Figures 1D and 1E are not new. For example, the constant protein noise as a function of mean protein expression is a known result of the two-state model of gene expression, e.g., see Equation (4) in Paulsson, Physics of Life Reviews 2005.

Yes, they may not be new, but we included these results for setting the baseline for comparison with simulation results that appear in the later part of the manuscript where we included translational bursting and ribosome demand in our models.

(6) In Figure 4C-D, it is unclear to me how the authors changed the mean protein expression if the translation initiation rate is a function of variation in mRNA number and other random variables.

The translation initiation rate varied from a basal translation initiation rate depending on the mRNA numbers and other variables. We changed the basal translation initiation rate to alter the mean protein expression levels. We have now elaborated the modelling section to incorporate these details in the revised manuscript (lines 404 to 412).

(7) If I understand correctly, the authors somehow changed the translation initiation rate to change the mean protein expression in Figures 4C-D. However, the authors changed the protein sequences in the experimental data of Figure 6. I am not sure if the comparison between simulations and experimental data is appropriate.

It is an important observation. Even though we changed the basal translation initiation rate to change the mean expression (Fig. 4C-D), we noted in the description of the model that the changes in the translation initiation rate were also linked to changes in the translation elongation rate (Fig. 3D). Thus, an increase in the translation initiation rate was associated with faster ribosome traversal through an mRNA molecule. This has also been observed in an experimental study by Barrington *et al.* (2023). Therefore, the models can also be expressed in terms of the translation elongation rate or ribosome traversal speed, instead of the translation initiation rate, and this modification will not change the results of the simulations due to interconnectedness of the initiation rate and the elongation rate.

**Reviewer #2 (Recommendations for the authors):**
Minor comments:(1) The discussion from lines 180 to 182 appears consistent with Figure 1E. It seems that the twostate model can already explain why the genes with high burst frequency and high protein synthesis rate showed a small protein noise. It is unclear to me the purpose of this discussion.

Yes, the results from Fig. 1E were from stochastic simulations, whereas the results discussed in the lines 191 to 193 (in the revised manuscript) were based on our analysis of experimental data that is shown in Fig. 2D.

(2) If I understand correctly, "translational efficiency" is the same as "protein synthesis rate" in this work. It would be helpful if the authors could keep the same notation throughout the paper to avoid confusion.

The protein synthesis rate per mRNA molecule is the best measure of translational efficiency, and we used the experimental data from Riba *et al.* (2019) for this purpose (line 99-100). Alternatively, we also used tRNA Adaptation Index (tAI) as a measure of translational efficiency, as the codon choice also influences the rate at which an mRNA molecule is translated (Tuller *et al*., 2010) (line 192).

(3) On line 227, does "higher translation rate" mean "higher translation initiation rate"? The same issues happen in a few places in this paper.

Corrected now (line 243 in the revised manuscript and throughout the manuscript).

(4) The discussion from lines 296 to 301 is unclear. It is not obvious to me how the authors obtained the conclusion that lowering translational efficiency would decrease the protein expression noise.

High translational efficiency will require more ribosomes and hence, will increase ribosome demand. If ribosome demand is the molecular basis of high expression noise for genes with bursty transcription and high translational efficiency, then we can expect a reduction in ribosome demand and a reduction in noise if we lower the translational efficiency. We have rephrased this section for clarity between the lines 334 and 339 in the revised manuscript.

(5) On line 324, should slower translation mean a shorter distance between neighboring ribosomes? One can imagine the extreme limit in which ribosomes move very slowly so that the mRNA is fully packed with ribosomes.

Slower translation or ribosome traversal rate would also lower the translation initiation rate (Barrington et al., 2023). Slower traversal of ribosomes reduces the chances of collision in case of transient slow-down of ribosomes due to occurrence of one or more non-preferred codons. We have now clarified this part in the lines 360 to 369 in the revised manuscript.

(6) The text from lines 423 to 433 can be put in Methods.

We have already added this part to the methods section (lines 900 to 910) and now minimize this discussion in the results section.

(7) The discussion from lines 128 to 130 is unclear, and the statement appears to be consistent with the two-state model (see Figure 1E). The meaning of "initial mRNA numbers" is also unclear.

An earlier study has proposed that essential genes in yeast employs high transcription and low translation strategy for expression, likely to maintain low expression noise in these genes and to prevent detrimental effects of high expression noise (Fraser et al., 2004). However, there has been no direct supportive evidence. Therefore, we were testing whether the differences in mRNA levels and translational efficiency of genes can lead to differences in protein noise through stochastic simulations. The discussion between the lines 130 and 132 in the revised manuscript summarises the results of the simulations.

Initial mRNA numbers - mRNA copy numbers that are present in the cell at the start of stochastic simulations. However, we have now changed it to ‘mRNA levels’ in the revised manuscript for clarity (line 131 in the revised manuscript).

(8) On line 212, is the translation initiation rate TL_init the same thing as beta_p in Figure 3A?

βp refers to the rate of protein synthesis, which is influenced by the translational burst kinetics as well as the translation initiation rate, whereas TLinit refers to the translation initiation rate. So, these parameters are related, but are not the same.